# Characterization of the NRAMP Gene Family in the Arbuscular Mycorrhizal Fungus *Rhizophagus irregularis*

**DOI:** 10.3390/jof8060592

**Published:** 2022-05-31

**Authors:** Víctor Manuel López-Lorca, María Jesús Molina-Luzón, Nuria Ferrol

**Affiliations:** Department of Soil Microbiology and Symbiotic Systems, Estación Experimental del Zaidín, CSIC, C. Profesor Albareda 1, 18008 Granada, Spain; victor.lopez@eez.csic.es (V.M.L.-L.); mariajesus.molina@eez.csic.es (M.J.M.-L.)

**Keywords:** arbuscular mycorrhizal fungi, iron, manganese, NRAMP transporters, *Rhizophagus irregularis*

## Abstract

Transporters of the NRAMP family are ubiquitous metal-transition transporters, playing a key role in metal homeostasis, especially in Mn and Fe homeostasis. In this work, we report the characterization of the NRAMP family members (*RiSMF1*, *RiSMF2*, *RiSMF3.1* and *RiSMF3.2*) of the arbuscular mycorrhizal (AM) fungus *Rhizophagus irregularis.* Phylogenetic analysis of the NRAMP sequences of different AM fungi showed that they are classified in two groups, which probably diverged early in their evolution. Functional analyses in yeast revealed that *RiSMF3.2* encodes a protein mediating Mn and Fe transport from the environment. Gene-expression analyses by RT-qPCR showed that the *RiSMF* genes are differentially expressed in the extraradical (ERM) and intraradical (IRM) mycelium and differentially regulated by Mn and Fe availability. Mn starvation decreased *RiSMF1* transcript levels in the ERM but increased *RiSMF3.1* expression in the IRM. In the ERM, *RiSMF1* expression was up-regulated by Fe deficiency, suggesting a role for its encoded protein in Fe-deficiency alleviation. Expression of *RiSMF3.2* in the ERM was up-regulated at the early stages of Fe toxicity but down-regulated at later stages. These data suggest a role for RiSMF3.2 not only in Fe transport but also as a sensor of high external-Fe concentrations. Both Mn- and Fe-deficient conditions affected ERM development. While Mn deficiency increased hyphal length, Fe deficiency reduced sporulation.

## 1. Introduction

Transition metals, such as iron (Fe) and manganese (Mn), are essential micronutrients necessary for the correct development and survival of all organisms. These metals play important roles in multiple biochemical processes, since they have structural roles in many proteins and act as enzyme cofactors. However, at high concentrations, they generate noxious reactive oxygen species (ROS) that are deleterious for growth and development [1,2]. Therefore, metal homeostasis must be strictly balanced at the cell and whole-organism level. To maintain cellular metal homeostasis, all organisms have evolved a series of mechanisms, including metal uptake, chelation, trafficking and storage systems [3]. Metal transporters play a major role in keeping appropriate metal concentrations in the different cellular compartments [4]. One of the most ubiquitous classes of metal transporters are the natural-resistance-associated macrophage proteins (NRAMPs) [5].

The NRAMP family represents an evolutionary-conserved strategy for acquiring and trafficking essential transition metals, such as Mn and Fe. All NRAMP transporters possess a highly conserved core of 10–12 transmembrane domains and two motifs that are essential for their transport function, the DPGN metal-binding domain at the first transmembrane domain and a high-conserved metal-transport motif in the cytoplasmic loop, between transmembrane domains 8 and 9 [6]. Members of the NRAMP family have been identified and characterised in many organisms, ranging from bacteria to mammals. The model yeast *Saccharomyces cerevisiae* has three homologs of this family in its genome: SMF1, SMF2 and SMF3 [7]. Smfp1 and Smfp2 were, initially, identified as extremely hydrophobic proteins that supress a lethal mutation in the yeast-mitochondrial-processing protein. Thereafter, they were shown to be Mn transporters that can, also, transport other divalent metal ions [8]. Smf1 is located at the plasma membrane and is responsible for Mn uptake, while Smf2p is located in intracellular vesicles and imports Mn for Mn-requiring enzymes. Smf3p localises at the vacuolar membrane and regulates vacuolar Fe transport [9]. While Smf1p and Smfp2 are induced by Mn deficiency and, to a lesser extent, by Fe deficiency [10], Smf3p is strongly induced under Fe starvation [9]. Orthologues of the SMF genes have been characterised in other fungi, including *Cryptococcus neoformans* [11], *Schizosaccharomyces pombe* [12,13] and *Aspergillus niger* [14], among others. Most fungal NRAMP family transporters have been involved in Mn homeostasis. For example, the SMF1 transporter of *Candida albicans* plays a role in Mn assimilation under alkaline conditions [15], and the high-affinity PsMnt transporter of the white-rot fungus *Phanerochaete sordida* is involved in cellular Mn accumulation under Mn-deficient conditions [16]. Other fungal NRAMP transporters, such as the plasma-membrane Fe-transporter EpNrampp of the dark endophyte *Exophiala pisciphila* [17] and the *Cryptococcus neoformans* Mn-transporter smf1 have been, also, involved in Cd tolerance [18], as their expression is down-regulated by Cd toxicity. In *A. niger*, deletion of the high-affinity Mn transporter DmtA leads to defects in germination and hyphal morphology [14]. However, NRAMP transporters in arbuscular mycorrhizal (AM) fungi remain uncharacterised.

AM fungi are obligate plant mutualistic microorganisms of the subphylum Glomeromycotina, within the Mucoromycota [19], that form a symbiotic association with most plant species [20]. The fungus colonises the root and forms highly branched structures, called arbuscules, inside the cortical cells. Outside the root, the fungus develops an extensive network of hyphae in the soil that can absorb nutrients beyond the depletion zone that develops around the roots. This extraradical mycelium (ERM) provides a new pathway to the plant, for the uptake of low-mobility nutrients in the soil, mainly P, N and some transition metals (Zn, Fe and Cu). In return, the plant provides sugar and lipids to the fungus [21,22]. This nutrient exchange takes place in the arbuscule-colonised cortical root cells. Besides enhancing nutrient uptake to their host plants, AM fungi provide increased tolerance against biotic and abiotic stresses, including drought, salinity or metal toxicity [23,24,25].

AM fungi play a crucial role in modulating plant metal acquisition, over a wide range of soil metal concentrations, as they increase plant metal acquisition in soils deficient in these elements but reduce metal uptake in contaminated soils [23,26]. Despite the importance of AM fungi for plant metal homeostasis, very few metal transporters have been characterised in these organisms. Up to now, just three components of the reductive iron uptake pathway (RiFTR1, RiFTR2 and RiFRE) and three copper transporters of the CTR family (RiCTR1-3) have been characterised in the model fungus *Rhizophagus irregularis* [27,28]. Iron uptake by the ERM starts with the reduction of Fe^3+^ to Fe^2+^ in the soil solution, by the plasma-membrane ferric reductase RiFre1. Then, Fe^2+^ is taken up by the plasma-membrane Fe permease RiFTR1 [27]. Although RiFTR2 function has not been determined, it has been proposed to play a role in Fe homeostasis, under Fe-limiting conditions. Regarding the *R. irregularis* Cu transporters, RiCTR1 mediates Cu uptake by the ERM from the environment, RiCTR2 is involved in mobilization of Cu vacuolar stores and RiCTR3a has been suggested to function as a Cu transceptor, involved in Cu tolerance [28]. Previous bioinformatics analysis of the *R. irregularis* genome identified four NRAMP family members (*RiSMF1*, *RiSMF2*, *RiSMF3.1* and *RiSMF3.2*) that remain uncharacterised [29]. At the plant side, up-regulation of NRAMP transporter genes has been observed in tomato *(LeNramp1 and LeNramp3* [30]) and alfalfa *(MsNramp1* [31]) mycorrhizal roots. However, *LeNramp1* and *LeNramp3* expression was down-regulated, when the symbiosis was established in a metal-contaminated soil, probably since the heavy-metal content is lower in mycorrhizal than in non-colonised roots [30].

The aim of this work was to characterise the *R. irregularis* NRAMP family members, in order to get further insights into the role of the NRAMP transporters in the AM symbiosis and into the mechanisms of the metal homeostasis in AM fungi. Data presented in this manuscript describe, for the first time, a Mn transporter in an AM fungus and show that *R. irregularis* uses various strategies for Fe uptake from the environment.

## 2. Materials and Methods

### 2.1. RiSMFs Sequence Analyses

Putative *RiSMFs* sequences were, previously, identified by Tamayo et al. [29]. Additional searches were performed in the filtered-model datasets of the *R. irregularis* isolates DAOM197198 v2.0 and A1, A4, A5, B3 and C2 v1.0 [32], deposited at the JGI (Joint Genome Institute) (https://genome.jgi.doe.gov/portal/, accessed on 10 April 2020), using as a query the previously identified RiSMF candidates. More recent datasets of the *R. irregularis* transcriptome were, also, searched to identify the NRAMP family members [33,34]. These candidate sequences were used to perform additional Blastp searches in JGI and NCBI (National Center for Biotechnology Information) databases, to find NRAMP homologs of other Glomeromycotina species. Primer3 (https://primer3.ut.ee/, accessed for the first time on 25 May 2020) was used to design the gene-specific primers. Homology and similarity comparisons were performed with SIAS (http://imed.med.ucm.es/Tools/sias.html, accessed on 10 June 2020). To search the protein-family domains and functional sites, InterProScan (http://www.ebi.ac.uk/Tools/InterProScan/, accessed on 19 April 2021) was used. Potential transmembrane domains were predicted using TOPCONS (https://topcons.cbr.su.se/, accessed on 20 April 2021). Structural models of the proteins were generated using the MyDomains tool of Prosite (https://prosite.expasy.org/mydomains/, accessed on 11 April 2021). The 3D models were predicted using the SWISS-MODEL software (https://swissmodel.expasy.org/, accessed on 12 May 2022), based on the crystal structure of *Eremococcus coleocola* divalent metal cation transporter MntH (ID: 5m87.1A). The alignment of putative amino acid sequences of NRAMPs from different fungi was performed with Clustal Omega (https://www.ebi.ac.uk/Tools/msa/clustalo/, accessed on 27 January 2022), and phylogenetic relationships were obtained applying the neighbour-joining method, implemented in MEGA-X [35]. FIMO (http://meme-suite.org/tools/fimo, accessed on 4 February 2022) was used to scan protein sequences, to find the Consensus Transport Motif (CTM).

### 2.2. RNA Sequencing Analyses

RNA-sequencing (RNAseq) data of laser microdissected *Medicago truncatula* cells, containing arbuscules (ARB) or IRM and of ERM grown in monoxenic cultures, were collected from the NCBI Gene Expression Omnibus database GSE99655. RNAseq data were analysed, as described by Zheng et al. [33]. Normalization of the counts was performed using DESeq2 tool of Galaxy (https://usegalaxy.org/, accessed on 17 May 2022).

### 2.3. Biological Materials and Growth Conditions

The AM fungal isolate used in this study was *Rhizophagus irregularis* DAOM197198. The fungal inoculum used for the monoxenic cultures and for the seedlings was obtained from monoxenic cultures. AM monoxenic cultures were established in bi-compartmental Petri dishes, containing solid M medium [36], to allow for separation of the root compartment (RC) from the hyphal compartment (HC) [37]. Cultures were started, by placing several segments of non-mycorrhizal *Agrobacterium rhizogenes* transformed-carrot (*Daucus carota*) roots and a piece of fungal inoculum containing ERM, fragments of mycorrhizal roots and spores, in the RC. Plates were incubated in the dark at 24 °C, until the other compartment was profusely colonised by the fungus and roots (6–8 weeks). Then, the RC was removed and refilled with fresh liquid M medium, and the ERM was allowed to colonise this compartment during 20 days (control plates).

Mn-deficient conditions were applied, by growing the ERM in liquid M medium without Mn, in the same conditions as the control plates. Fe-deficient conditions were induced, by exposing the ERM grown in liquid M medium without Fe to 0.5 mM ferrozine for 3 days, as described by Tamayo et al. [27].

Two experiments were set up to assess the effect of Fe toxicity on *RiSMF* expression: a short- and a long-term experiment. In the short-term experiment, the ERM grown in liquid M media control was exposed for 16 h to 45 mM EDTAFe (III) sodium salt. In the long-term experiment, the ERM was grown in liquid M media, supplemented with 45 mM Fe, for 20 days. ERM from the different treatments were collected using tweezers, washed with distilled water, dried on filter paper, frozen in liquid N and stored at −80 °C until used.

*R. irregularis* ERM was, also, collected from chicory mycorrhizal plants grown in the in vivo whole-plant bidimensional experimental system [38]. Chicory (*Cichorium intybus* L.) seeds were surface sterilised and germinated in autoclaved HCl-washed sand under sterile conditions. Sand was washed before sterilisation with 0.03 M HCl to eliminate metal traces and then rinsed with distilled water until pH 7. Seedlings were allowed to grow in a growth chamber under 16 h light at 24 °C and 8 h dark at 20 °C for two weeks. Then, seedlings were transferred to pots containing 50 mL of sterilised HCl-washed sand and inoculated with 3 mL of AM fungal inoculum, obtained in monoxenic cultures and containing 1650 spores. To prevent substrate desiccation, pots were introduced in sun-transparent bags (Sigma-Aldrich, B7026, St. Louis, MO, USA). Plants were watered with half-strength (0.5X) Hoagland nutrient solution and grown for one month at 24 °C day/20 °C night and in a 16 h light photoperiod. To induce Mn- or Fe-deficient conditions, plants were watered with Hoagland solution without MnCl_2_ or EDTA Fe (III) sodium salt, respectively. Then, the root system of each plant was washed with distilled water, wrapped in a nylon net (41 µm mesh, Millipore NY4100010, Merk, Darmstadt, Germany) and sandwiched between two 13 cm membranes of mixed-cellulose esters (0.45 µm pore diameter size, MF-Millipore HAWP14250, Merk, Darmstadt, Germany). The sandwiched root systems were placed into 14 cm diameter petri plates, having a hole on the edge to allow shoot growth, and filled with sterile HCl-washed sand. Petri plates containing plants were sealed with parafilm, wrapped with aluminium foil, placed into sun-transparent bags and maintained in a growth chamber. Plates were incubated for another month, until ERM colonised the Millipore membrane. Finally, ERM spreading from the nylon net onto the membranes was collected with tweezers. At harvesting, plant biomass was determined, by measuring shoot and root fresh weights. ERM and root systems were frozen in liquid N and stored at −80 °C until used. An aliquot of the roots was separated, to estimate mycorrhizal colonization.

The *Saccharomyces cerevisiae* yeast strains used in this study were *smf1*Δ, a single mutant lacking the plasma-membrane transporter Smf1 [39], and *fet3*Δ*fet4*Δ, a double mutant defective in both high- and low-affinity iron transport [40]. Yeast cells were grown on YPD or minimal synthetic dextrose (SD) medium, supplemented with the appropriate amino acids.

### 2.4. RNA Extraction and Gene Expression Analyses

Total RNA was extracted from ERM developed in monoxenic and sandwich cultures, using the RNeasy Plant Mini Kit (Qiagen, Hilden, Germany), in accordance with the instructions of the manufacturer. Total plant RNA was isolated from chicory roots, by using the phenol/SDS method, followed by LiCl precipitation [41]. RNAs were treated with DNase, using the RNA-free DNAse set (PROMEGA, Madison, WI, USA), in accordance with the protocol of the manufacturer. cDNAs were synthetised from 1 µg of total DNase-treated RNA, in a 20 µL reaction using Super-Script IV Reverse Transcriptase (Invitrogen, Carlsbad, CA, USA), in accordance with the instructions of the manufacturer.

*RiSMFs* gene expression was analysed by real-time RT-PCR, using a QuantStudio 3 (Applied Biosystem, Waltham, MA, USA), in the synthetised cDNAs. Each 12 µL reaction contained 1 µL of a 1:10 dilution of cDNA, 0.25 µL each primer and 6 µL iTaq (Bio-Rad, Hercules, CA, USA). Specificity of the different primer sets (Appendix A) was analysed by PCR amplification of *R. irregularis* and chicory cDNA. The real-time RT-PCR program consisted of an initial incubation at 95°C for 30 s, followed by 40 cycles of 95 °C for 15 s, 60 °C for 30 s and 72 °C for 30 s, where the fluorescence signal was measured, and a final step with a heat-dissociation protocol to check the specificity of the PCR-amplification procedure. The efficiency of the different primer pairs was determined, through a real-time RT-PCR on several dilutions of cDNA. The results obtained for the different treatments were standardised to *RiEF1**α*. Real-time RT-PCR determinations were carried out on at least three independent biological samples, from three replicate experiments. Real-time RT-PCR reactions were performed, at least three times for each biological sample, with the threshold cycle (Ct) determined in triplicate. Relative expression level was calculated using the 2-^Δ^^Ct^ method, and the standard error was computed from the average of the ΔCt values, for each biological sample.

### 2.5. Heterologous Expression

The coding regions of the *R. irregularis* NRAMP genes were amplified from ERM cDNA by PCR, using the corresponding primers pairs (Appendix A) and cloned into the yeast-expression-vector pDRf1-GW, using the Gateway technology (Invitrogen, Carlsbad, CA, USA). The respective full-length cDNA sequences were flanked with the sequences attB1 and attB2, to be recognised by the BP Clonase enzyme. PCR products were, first, cloned into pDONr221 and, then, cloned into the yeast expression vector pDRf1-GW, in accordance with the instructions of the manufacturer. All constructs were verified by sequencing. Although a PCR band of the expected size was obtained for *RiSMF2*, all attempts we did to clone it failed. *S. cerevisiae* mutant strains *smf1*Δ and *fet3*Δ*fet4*Δ were transformed with the pDRf1-RiSMF1, pDRf1-RiSMF3.1 or pDRf1-RiSMF3.2 constructs or with the empty vector (negative control), using a lithium-acetate-based method [42]. Yeast transformants were selected in SD medium, by uracil autotrophy. For drop tests, yeast transformants were harvested by centrifugation, from a liquid culture grown to the exponential phase in SD medium without uracil, washed three times with milli-Q H_2_O and adjusted to a final OD_600_ of 1. Then, 5 µL of serial 1:10 dilutions were spotted on the corresponding selective medium. *fet3*Δ*fet4*Δ transformants were spotted onto SD without uracil pH 3.5 supplemented with 100 µM FeCl_3_ or SD without uracil pH 5.4. The transformed *smf1*Δ strain were spotted onto SD without uracil plates supplemented with 50 mM MES pH 6, 6.25 mM EGTA and supplemented or not with 500 μM MnSO_4_.

### 2.6. RiSMFs Subcellular Localization

Subcellular localization of RiSMFs was performed, with carboxi-terminal fusions of these genes, to the enhanced green fluorescent protein (eGFP) in the *S. cerevisiae* mutant *smf1*Δ. *RiSMF1*, *RiSMF3.1* and *RiSMF3.2* open-reading frames were PCR amplified, without their stop codon, using the corresponding primers (Appendix A), and cloned into the yeast expression vector pGWFDR196, using the Gateway technology (Invitrogen). Mutant yeasts were transformed with the pGWFDR196:RiSMF constructs or with the empty vector (negative control). Functionality of the GFP-fusion proteins was tested in complementation assays, as previously described. For the localization assays of the RiSMFs, yeast cells were grown to the exponential phase in liquid SD without uracil, washed with distilled water and visualised with a Leica DMi8 microscope. GFP-tagged proteins were observed, using a 483–501 nm excitation filter and a 512–548 nm emission filter.

### 2.7. Histochemical Staining of Fe

Spores developed in monoxenic cultures in control plates and in M medium supplemented with 45mM EDTA-Fe (III)Na were stained with the PERLS reagent, for Fe detection. Briefly, spores were harvested with tweezers, transferred to a microtube, rinsed with milli-Q H_2_O and placed on a slide. Spores were incubated with an equal volume of 4% (*v*/*v*) HCl and 4% (*w*/*v*) K-ferrocyanide (Perls stain solution), for 2 min at room temperature. Negative stain controls were prepared, by incubating a set of spores with HCl. Spores were washed with dH_2_O and bright-light visualised with a Nikon SMZ1000 binocular.

### 2.8. Mycorrhizal Colonization and ERM Development

Mycorrhizal colonization was estimated in an aliquot of the root system of each chicory plant stained with trypan blue, following the Trouvelot method [43]. Expression of the *R. irregularis* elongation factor 1α (*RiEF1**α*) in roots was, also, used as a molecular marker of the abundance of the AM fungus. ERM development in the in vivo whole-plant bidimensional experimental system was assessed, by staining the ERM with trypan blue on the Millipore membranes. Hyphal length and spore density were calculated, under a binocular microscope, a Nikon SMZ1000, in three 1 cm^2^ squares. The number of spores was counted over the three squares, and the hyphal length (L) was estimated by the number of hyphal intersections, with the grid (N) using the modified Newmann’s formula [44], L(cm) = N × (11/14) × grid unit(cm), where 11/14 is the ideal size of the grid side, to obtain the estimation in cm and grid unit = 1 cm. Three replicate plates were considered per treatment.

### 2.9. Statistical Analyses

Statgraphics Centurion XVI software was used for the statistical analysis of means and standard error determinations. For comparison of treatments, a one-way ANOVA followed by a Duncan test (*p* < 0.05) were performed.

## 3. Results

### 3.1. The NRAMP Family of R. irregularis

The genome of *R. irregularis* harbours four genes putatively encoding NRAMPs that were previously named according to their *S. cerevisiae* homologues [29]. Their full-length cDNAs contain open reading frames of 1422–1665 nucleotides. The deduced amino acid sequences of *RiSMF1*, *RiSMF2*, *RiSMF3.1* and *RiSMF3.2* comprise 554, 473, 501 and 525 amino acid residues, respectively, and have a calculated molecular mass of 52.77–61.07 kDa. The amino terminus of all RiSMFs faces the cytosol. RiSMF1 and RiSMF2 have 11 transmembrane domains, while RiSMF3.1 and RiSMF3.2 have 12 (Figure 1a,d). They all contain the NRAMP family domain DPGN, which is essential for metal binding, and the consensus transport motif GQSSTITGTYAGQY(/F)V(/I)MQGFLD(/E/N) in the cytoplasmic loop, between transmembrane domains 8 and 9 (Figure 1a,b). Alignment of the full-length cDNAs with the genomic DNA sequences revealed the presence of three introns in *RiSMF1*, one in *RiSMF2*, five in *RiSMF3.1* and four in RiSMF3.2 (Figure 1c). Sequence similarity among the deduced amino acid sequences, of the different RiSMFs, ranges from 38% to 60%, with RiSMF3.1 and RiSMF3.2 showing the highest degree of similarity (60%). The *R. irregularis* NRAMP proteins show high amino acid sequence conservation, with NRAMPs of *S. cerevisiae* (26 to 38%) and of other kingdoms, such as the NRAMP proteins of the bacteria *Paenibacillus* sp. (30 to 40%), *Arabidopsis thaliana* (28 to 42%) and *Caenorhabditis elegans* (30 to 40%) (Appendix A).

A phylogenetic analysis of the NRAMP sequences of different kingdoms revealed that the *R. irregularis* NRAMPs were grouped in two different clades (Group I and Group II). Group I clusters RiSMF1, RiSMF2 and other Glomeromycotina sequences within the branch of fungal NRAMPs. RiSMF3.1, RiSMF3.2 and other Glomeromycotina sequences were clustered in a branch placed between the two groups of plant NRAMP sequences (Group II). Four NRAMP gene sequences were identified in the genome of AM fungi, belonging to the order Glomerales, whereas two were found in the genome of *Gigaspora rosea* and two in the genome of *Diversispora epigea*. All species analysed harbour NRAMP genes in Groups I and II (Figure 2).

### 3.2. Functional Characterization of RiSMFs in Yeast

Since AM fungi cannot be genetically manipulated, functionality of the RiSMFs was tested trough heterologous complementation in yeast. As a first step to investigate their function, the ability of RiSMF1, RiSMF3.1 and RiSMF3.2 to complement the phenotype of a yeast strain disrupted in the *SMF1* gene (*smf1*Δ) was tested. Unfortunately, functionality of RiSMF2 could not be analysed, as we were unable to clone its full-length cDNA into the yeast-expression vector. Yeast smf1, a NRAMP homologue, encodes a plasma-membrane Mn transporter with a broad metal specificity. *smf1*Δ cells fail to grow on SD medium, containing high concentrations of the divalent chelator EGTA, an effect that is relieved by Mn [45]. Expression of *RiSMF3.2*, under the control of the yeast PMA1 promoter, complemented the phenotype of the *smf1*Δ cells. However, a strain transformed with the empty vector pDRF1-GW, or with the pDRf1:RiSMF1 or pDRf1:RiSMF3.1 plasmids, could not grow on SD media supplemented with 6.25 mM EGTA (Figure 3a). These data demonstrate that *RiSMF3.2* cDNA encodes a functional protein that mediates Mn transport.

Subsequently, we tested whether the RiSMFs were able to complement the yeast Fe transport mutant *fet3*Δ*fet4*Δ. This strain is defective in both low- and high-affinity Fe uptake systems and requires high Fe concentrations in its medium of growth as well as a more acidic environment, to increase this micronutrient availability [46]. Therefore, the positive controls were prepared, by growing the *fet3*Δ*fet4*Δ cells transformed with the empty vector pDRF1-GW or with the pDRf1:RiSMF1, pDRf1:RiSMF3.1 or pDRf1:RiSMF3.2 plasmids on SD-URA medium pH 3.5, supplemented with 100 μM FeCl_3_. As it was observed for the *smf1*Δ mutant, only RiSMF3.2 restored the ability of *fet3*Δ*fet4*Δ to grow on SD medium (pH 5.4) that was not supplemented with Fe (Figure 3b). These data provide additional evidence that RiSMF3.2 is a metal transporter. Therefore, RiSMF3.2 is a functional homologue of the yeast plasma-membrane smf1 transporter, which mediates transport of Mn and Fe.

To determine subcellular localization of RiSMF3.2 in the heterologous system and to assess whether the failure of RiSMF1 and RiSMF3.1 to complement the phenotypes of the *smf1*Δ and *fet3*Δ*fet4*Δ cells might be because the proteins were located in an intracellular membrane, C terminal GFP-tagged versions of these proteins were expressed in the *smf1*Δ strain, under the control of the PMA1 promoter. Functionality of the RiSMF-fusion proteins was assessed, before their visualization by fluorescence microscopy. None of the RiSMF-GFP fusion proteins complemented the mutant phenotype of the *smf1*Δ strain (Appendix A). Cells expressing the RiSMF-GFP fusion proteins showed a general cytosolic fluorescent signal, similar to the one showed by control cells expressing the soluble GFP (Appendix A). These data suggest a degradation of the misfolded fusion proteins in the proteasomes, a phenomenon, previously, observed in yeast-membrane protein-expression assays [47].

### 3.3. RiSMF Genes Are Differentially Expressed in Various Fungal Structures

To try to understand the biological function of the *RiSMF*s, their expression levels in various fungal structures were investigated, by using recent RNA-Seq data retrieved from ERM grown in monoxenic cultures, laser-microdissected *Medicago truncatula* cortical cells containing arbuscules and laser-microdissected *M. truncatula* root cells containing IRM [33,34]. The four *R. irregularis SMF* genes were expressed in all fungal structures (Figure 4). *RiSMF2* was highly expressed in the IRM and in the arbuscules. *RiSMF1* was the gene displaying the higest expression level in the ERM, followed by *RiSMF3.2*.

To verify these gene expression patterns, *RiSMF*s expression was analysed by RT-qPCR in ERM and IRM, grown in association with chicory plants in the in vivo whole-plant experimental sandwich system, and in ERM grown in monoxenic cultures. Transcripts of the four *R. irregularis* NRAMP family members were detected, both in the ERM and IRM (Figure 5). Interestingly, *RiSMF1* and *RiSMF2* expression levels were different in ERM grown in monoxenic cultures and in the whole-plant sandwich culture system, which might be due to the different physiological and developmental stages of both ERM. The ERM grown in monoxenic cultures was developed in association with a root organ culture lacking the plant stem and leaves, which might affect development of the fungus. As observed in the RNAseq analyses, *RiSMF1* was the most highly expressed isoform in the ERM, and no significant differences were observed between *RiSMF3.1* and *RiSMF3.2* expression in ERM and IRM. While *RiSMF2* was the most highly expressed gene in the IRM in the RNAseq analysis, the RT-qPCR gene expression analyses showed that *RiSMF1* was the most abundant gene in chicory mycorrhizal roots. This discrepancy might be because different host-plant species were used in both experiments. While *Medicago* was used in the RNAseq analyses, chicory was used in the RT-qPCR assays.

### 3.4. Mn Starvation Regulates RiSMF1 and RiSMF3.1 Expression

To gain further information about the potential roles of the RiSMF isoforms, their expression levels were analysed by real-time RT-PCR in ERM and IRM grown in association with chicory plants in the in vivo sandwich system, under Mn-sufficient and Mn-limiting conditions. Mn deficiency decreased *RiSMF1* transcript levels in the ERM and increased *RiSMF3.1* expression in the IRM (Figure 6a).

The effect of Mn availability on *RiSMF*s expression was also analysed in RNAs isolated from ERM grown in vitro in monoxenic cultures with and without Mn. Expression of none of the *RiSMF* genes was significantly affected by Mn deficiency in the ERM grown in the in vitro culture system (Figure 6b).

### 3.5. Impact of Fe Availability on RiSMFs Expression

Taking into account that fungal NRAMPs are regulated by Fe and that *RiSMF3.2* has an Fe transport function, the effect of Fe availability on *RiSMF*s expression was assessed, to get further insights into their roles in *R. irregularis*. Firstly, the effect of Fe starvation was analysed in ERM and IRM grown in the in vivo sandwich system, under control and Fe-deficient conditions. A two-fold increase in *RiSMF1* expression was observed in the ERM developed in plants watered with a nutrient solution without Fe. However, *RiSMF2*, *RiSMF3.1* and *RiSMF3.2* transcript levels were not significantly affected, neither in ERM nor IRM (Figure 7a). Up-regulation of *RiSMF1* by Fe deficiency was, also, observed when the ERM was grown in vitro in monoxenic cultures (Figure 7b).

To assess whether the *R. irregularis* NRAMPs could play a role on ERM Fe tolerance, *RiSMF*s expression was analysed, in ERM grown in monoxenic cultures and exposed for 16 h to 45 mM Fe and in ERM grown for 20 days in a medium supplemented with 45 mM Fe. Transcript levels of *RiSMF1*, *RiSMF2* and *RiSMF3.1* were not significantly affected by Fe toxicity. However, expression of *RiSMF3.2* was transiently up-regulated by Fe toxicity. A 2.5-fold increase was observed 16 h after Fe addition, while a three-fold decrease was observed when the ERM was grown for 20 days in the presence of 45 mM Fe (Figure 8a).

Spores formed under Fe-toxic conditions showed a dark brown colour, suggesting that they might have a high Fe content. To test this hypothesis, spores were incubated in the Perls Prussian blue staining solution (HCl and K-ferrocyanide), which specifically stains Fe labile iron in biological tissues, by forming a blue precipitate of ferric ferrocyanide (Prusian blue) [48]. After incubation in the staining solution, the spores turned blue. A blue precipitate was detected in the internal cell wall layers of the crushed spores developed under Fe toxicity, but not in spores grown under control conditions (Figure 8b–i). Moreover, a blue precipitate was detected in the ERM and spore cytoplasm. These data are indicative of Fe accumulation in the fungal spores and ERM.

### 3.6. Impact of Mn and Fe Deficiency on R. irregularis Development

Taking into account that, in fungi, Mn deficiency impacts hyphal development and sporulation [14], the effect of Mn deficiency on the development of the ERM and IRM of *R. irregularis* was evaluated, in the chicory plants grown in the in vivo system. As revealed by Trouvelot quantification of trypan blue-stained roots and by expression analysis of the *R. irregularis EF1**α* gene, mycorrhizal-colonization levels were not affected by Mn deficiency (Table 1). Spore density, measured in membranes of the sandwich cultures, was not affected either, but an increase in hyphal length was observed under Mn-deficient conditions (Table 1).

As observed for the Mn-deficiency experiment, IRM development in chicory roots was not affected by Fe deficiency. However, sporulation was significantly reduced, while hyphal length was not significantly affected, when the ERM was developed under Fe-limiting conditions (Table 2).

## 4. Discussion

A previous genome-wide analysis of metal transporters in *R. irregularis* revealed the presence of four gene sequences, *RiSMF1*, *RiSMF2*, *RiSMF3.1* and *RiSMF3.2*, putatively encoding transporters of the NRAMP family [29]. Mining of the more recent *R. irregularis* genome and transcriptome databases confirms that the *R. irregularis* NRAMP family is composed of four members. In this work, we functionally characterised *RiSMF3.2* and analysed gene expression patterns of the *R. irregularis* NRAMP family members. Our data indicate that the *RiSMF* genes are differentially regulated by Mn and Fe availability, and that *RiSMF3.2* encodes a functional Mn and Fe transporter.

The obligate biotrophic and multinucleate nature of AM fungi prevents the use of the most common strategies to investigate the functionality of a gene of interest. Recent advances have been made to assess the functionality of AM fungal genes, by host-induced gene silencing or virus-induced gene silencing [49,50]. However, these techniques are used to investigate the function of genes highly expressed in the IRM and have not always been successfully applied. To bypass this technical constrain, we tried to assess the function of the *RiSMF* gene products in yeast. Unfortunately, we could only determine the transport function of *RiSMF3.2* in the heterologous system.

Despite only *RiSMF3.2* presented a metal transport activity in yeast, the four *R. irregularis* NRAMP sequences contain all the structural features of NRAMP proteins. In fact, they contain the conserved transmembrane motif GQSSTITGTYAGQY(/F)V(/I)MQGFLD(/ E/N) and the DPGN motif characteristic of the NRAMP family [6]. Numerous mutational studies have shown that both domains are essential for the metal-transport activity of NRAMP transporters [51]. For example, conservative substitutions of the aspartate (D) and asparagine (N) residues impaired metal binding of the ScaNramp of *Staphylococcus capitis* [52] and eliminated metal transport in human NRAMP2 [53] and *Ecoli*Nramp [54]. Thus, this motif in the *R. irregularis* NRAMP proteins should, also, play a role in metal binding.

The phylogenetic analysis revealed that the Glomeromycotina NRAMP sequences are divided in two subfamilies, a Group I belonging to the clade grouping fungal NRAMP sequences, and a Group II that is independent from the clusters formed by animal, plant, fungal and bacterial sequences. Group II is more closely related to known plant NRAMP sequences than to fungal sequences. In addition to the divergence in primary amino acid sequence between genes from Group I and II, they display some differences in gene organization. Members of Group II are more fragmented and have three–four introns, located at conserved positions at the carboxi-terminus. This sequence divergence suggests an early evolutionary separation between the two groups of Glomeromycotina NRAMPs. The observation, that all AM fungal species analysed have NRAMP genes from Group I and Group II, suggests that both groups are required for proper metal homeostasis in AM fungi. Further studies are required, to determine if proteins of the two groups display different transport functions.

The finding that RiSMF3.2 reverts the mutant phenotype of the *smf1*Δ and *fet3*Δ*fet4*Δ strains, lacking, respectively, the high-affinity Mn transporter smf1 and the iron uptake systems, indicates that *RiSMF3.2* encodes a NRAMP transporter that mediates Mn and Fe transport. Although subcellular localization of RiSMF3.2 could not be demonstrated in yeast, it is the orthologue of the plasma-membrane smf1 transporter. The other members of the *R. irregularis* NRAMP family could not be characterised in the heterologous system because, as revealed by the yeast localization assays, they were not expressed in the yeast membranes, most likely as a consequence of an artefact of the heterologous system. Although the principles of targeting seem to be conserved between organisms, the heterologous proteins may lack the sequences required for targeting to the correct compartment in the cell, so problems regarding the correct folding and targeting of the heterologous expressed proteins can occur [55].

RiSMF1 function could not be determined, but it seems to be involved in Mn and Fe homeostasis, as its expression levels in the ERM are regulated by Mn and Fe availability. The contrasting expression patterns of *RiSMF1* in the ERM grown in the in vivo and in vitro cultures systems under Mn deficient conditions may be because the ERM grown in monoxenic cultures is not Mn harvested. Since the culture medium of the root compartment contains Mn, it is possible that Mn is transferred from the IRM to the ERM in the monoxenic cultures. Down-regulation of *RiSMF1* by Mn deficiency is striking, as high-affinity metal transporters are expected to be up-regulated under metal deficient conditions. However, unexpected regulation patterns of NRAMP genes by Mn have been observed in various organisms. For example, expression of the *Aspergillus oryzae AoNramp1* gene increases under Mn toxicity [56] and transcript levels of the cucumber *CsNRAMP1*, *CsNRAMP4* and *CsNRAMP5* genes decrease under Mn deficiency [57]. How Mn availability affects NRAMP gene expression remains to be investigated. As it has been described for the yeast smf1p and smf2p transporters, the *RiSMFs* could, also, be regulated at the post-translational level by Mn [10].

Up-regulation of *RiSMF1* expression in the ERM, under Fe-limiting conditions, suggests a role for its encoded protein in Fe-deficiency alleviation. Since the RiSMF1 subcellular location could not be determined in the heterologous system, its role in Fe homeostasis could either be due to its capacity to increase Fe uptake from the environment or to mobilise the vacuolar Fe stores under Fe-limiting conditions. Although *RiSMF3.2* transcript levels were not regulated by Fe deficiency, the yeast-complementation assays revealed that it encodes a plasma-membrane Fe transporter of the NRAMP family involved in Fe uptake. Recent work by Tamayo et al. [27] has identified two *R. irregularis* Fe permeases (RiFTR1 and RiFTR2) involved in Fe homeostasis. RiFTR1 is involved in Fe acquisition by the plasma membrane and RiFTR2 in Fe homeostasis under Fe-limiting conditions. These data, altogether, suggest that *R. irregularis* uses various strategies to increase Fe uptake from the environment, the plasma-membrane RiFTR1 and RiSMF3.2 transporters. Similarly, multiple systems operate at the cell surface for Fe uptake in *S. cerevisiae*, the high-affinity Fe transporters Ftr1p and smfp1 and the low-affinity transporter Fet4p [8,46,58]. Additional studies at the protein level are needed to understand whether RiSMF1 mediates Fe uptake from the environment or mobilises the Fe vacuolar stores and the relative contribution of the different Fe transporters to Fe uptake by the ERM. As expected for an uptake plasma-membrane transporter, *RiSMF3.2* expression levels decreased when the ERM was grown in a media containing 45 mM Fe for 20 d. A similar expression pattern was observed by Tamayo et al. [27], for the Fe transporters RiFTR1 and RiFTR2, under these experimental conditions. Yeast NRAMP Smf1p, the orthologue of RiSMF3.2, is also down-regulated by metal toxicity [59]. Transcriptional down-regulation of the *R. irregularis* proteins, mediating Fe transport into the cytosol, will limit uptake of toxic levels of Fe by the ERM. However, the effect of Fe toxicity on *RiSMF3.2* expression, 16 h after the addition of 45 mM Fe, resulting in enhanced transcript levels, was striking. This difference implies that the contribution of RiSMF3.2 at the early stages of Fe toxicity would be increased Fe uptake, leading to an increased Fe toxicity. Alternatively, it could be hypothesised that RiSMF3.2 could act as a sensor of high external Fe concentrations, to activate the signalling cascades involved in Fe tolerance. In fact, nutrient sensing in fungi can be mediated by transceptors, which are proteins with both transport and receptor functions [60,61]. In the absence of a methodology to silence AM fungal genes in the ERM, it is not possible to understand the biological significance of this transient increase in *RiSMF3.2* mRNA levels.

Gene expression of the *RiSMF*s in the IRM reveals the importance of keeping both Mn and Fe homeostasis during the *in planta* phase of the fungus. The finding that *RiSMF2* was the most highly expressed gene in IRM and arbuscules collected from *Medicago* roots, while *RiSMF1* was the gene displaying the highest expression levels in carrot mycorrhizal roots, indicates that expression of these genes is regulated by the plant genotype and/or the experimental conditions. Host-dependent expression of a subset of *R. irregularis* secreted proteins has been, also, reported [33]. Further studies are required to determine the host cues regulating *RiSMF1* and *RiSMF2* expression in the IRM, and if they are involved in metal uptake from the apoplast of the symbiotic interface. Up-regulation of *RiSMF3.1* expression by Mn deficiency in the IRM suggests that, under Mn starvation, the fungus needs to increase its Mn cytosolic content, in order to provide it to the Mn-requiring enzymes, such as the mitochondrial Mn superoxide dismutase and the Golgi-located enzymes involved in the glycosylation of secretory proteins [62]. In fact, the yeast smf2p transporter has been shown to be a central player in Mn trafficking to the mitochondria and other cellular sites [63]. Taking into account that expression of the *RiSMF* genes is not affected by Fe deficiency in the IRM and the high expression levels reported for the high-affinity transporter RiFTR1 in the IRM [27], it is likely that RiFTR1 is the major player in Fe homeostasis, in the structures the fungus develops in the root.

Based on data presented in this manuscript and on previous reports of the *R. irregularis* Fe uptake systems [27], we propose a model for Mn and Fe transport in the AM fungus *R. irregularis* (Figure 9). Our gene-expression and functional analyses in yeast strongly suggest that RiSMF3.2 is involved in Mn and Fe uptake by the ERM from the soil solution and by the IRM from the apoplast of the symbiotic interface. The plasma-membrane Fe transporter RiFTR1 also contributes to Fe uptake in both fungal structures [27]. In depth analyses of the cellular function of the identified NRAMP transporters RiSMF1, RiSMF2 and RiSMF3.1 in different fungal structures is required, to better understand whether they also contribute to Mn and Fe uptake, or if they are involved in Mn trafficking or in mobilization of the metal vacuolar stores.

Numerous studies have shown that AM fungi increase plant acquisition of the essential metals Zn, Cu and Fe; however, information about the role of the symbiosis in plant Mn nutrition and on the underlying mechanisms is scarce. While a few studies have shown that Mn uptake is higher in mycorrhizal plants [64,65], it has been, repeatedly, reported that Mn acquisition decreases in mycorrhizal plants [66]. AM fungi have been shown to reduce the number of Mn-reducing bacteria [67] or increase the number of Mn-oxidizing bacteria in the rhizosphere [68], decreasing indirectly Mn availability. Nevertheless, the observed hyphal length observed under Mn-deficient conditions indicates that, under these conditions, the fungus explores a higher volume of soil, which will increase the nutrient-uptake effectiveness of the mycorrhizal root. Under field conditions, Mn uptake by mycorrhizal roots may depend on which of the two functions (Mn availability in the mycorrhizosphere or volume of soil exploited by the mycorrhizal root) prevails in the soil.

Regarding the effect of Mn on the developmental partner of the ERM, the increased hyphal length observed, when the fungus grows in the absence of Mn, agrees with previous observations for other nutrients [69,70]. As proposed by Bago et al. [69] and Olsson et al. [70], this growth pattern is, probably, designed to explore and exploit more efficiently the growth medium. An effect of Mn deficiency on hyphal development has been, also, reported in *Aspergillus niger* [14]. Although Fe deficiency did not affect hyphal length, it decreased sporulation. Fe is an essential micronutrient that is a cofactor of numerous enzymes, thanks to its ability to easily accept and release electrons [71,72]. Therefore, inhibition of sporulation might be as a consequence of the inhibition of the activity of the enzymes required for spore formation, when the fungus is grown in media lacking Fe. Previous studies have shown that AM fungal-spore formation is affected by nutrient availability [64]. Detection of Fe in the spores developed under Fe toxicity agrees with previous observations for other metals, such as Cu, Zn and Cd [73,74,75], and supports the hypothesis that a survival strategy of AM fungi in metal-contaminated environments is to accumulate the excess metal in some spores of the fungal colony. Iron accumulation in the fungus will reduce plant Fe availability, which will explain, at least partially, the improved performance of mycorrhizal plants, in soils affected by iron mining tailing [76,77].

## 5. Conclusions

This manuscript describes, for the first time, characterization of the NRAMP family members, the *RiSMF* genes, in an AM fungus. The *R. irregularis* SMF genes are expressed both in the ERM and IRM and are differentially regulated by environmental Fe and Mn. *RiSMF3.2* encodes a protein mediating Mn and Fe transport from the environment, being the first Mn transporter reported in an AM fungus. These data indicate *R. irregularis* uses various strategies to increase Fe uptake from the environment: the previously identified plasma-membrane Fe permease RiFTR1 and the RiSMF3.2 NRAMP transporter. Further studies are required to understand the relative contribution of these transporters to Fe uptake by the IRM and the ERM and to elucidate the role of the other members of the *R. irregularis* NRAMP family.

## Figures and Tables

**Figure 1 jof-08-00592-f001:**
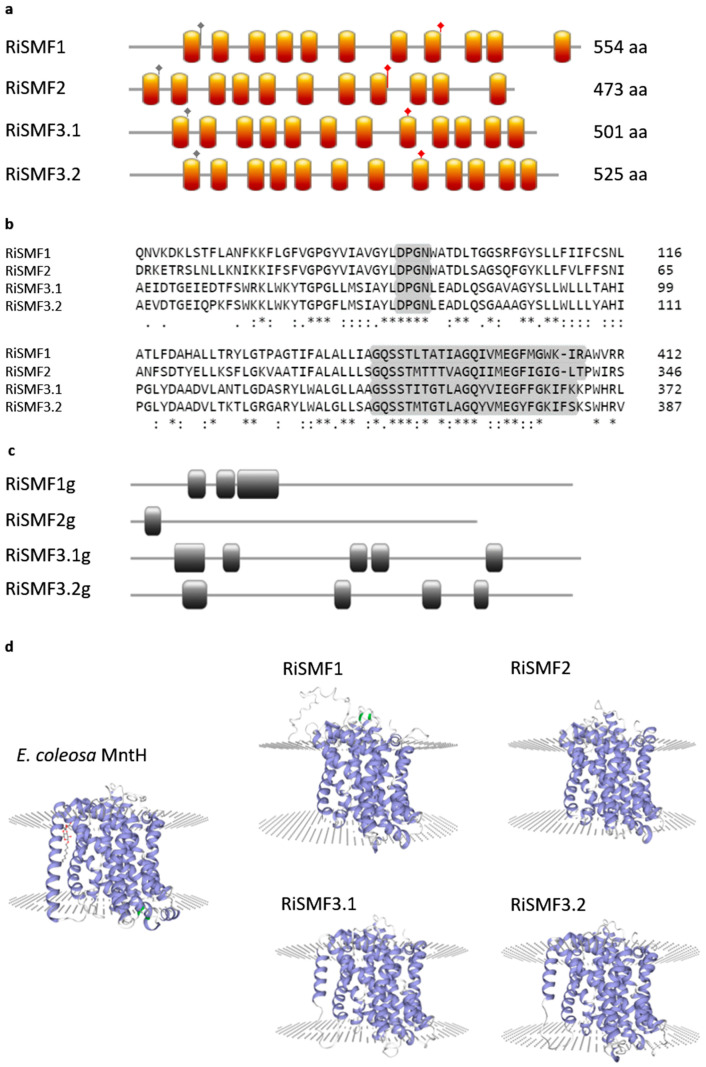
(**a**) Representation of the structure of *R. irregularis* NRAMP transporters. Orange boxes represent transmembrane domains, grey diamonds show the position of the DPGN motif and red diamonds show the position of consensus transport motif. (**b**) Multiple Sequence Alignment of the four protein sequences, with the DPGN and consensus transport motif highlighted. * indicates positions which have a single, fully conserved residue; indicates conservation between groups of strongly similar properties; indicates conservation between groups of weekly similar properties, (**c**) Exon/intron organization of *RiSMF* genes. Exons and introns are represented by grey lines and grey boxes, respectively. (**d**) Predicted 3D structure of RiSMF transporters. Tertiary structures were predicted by SWISS-MODEL software, based on the template of *E. coleosa* MntH (ID: 5m87.1A). α-helices and β-lamina are represented in blue and green, respectively.

**Figure 2 jof-08-00592-f002:**
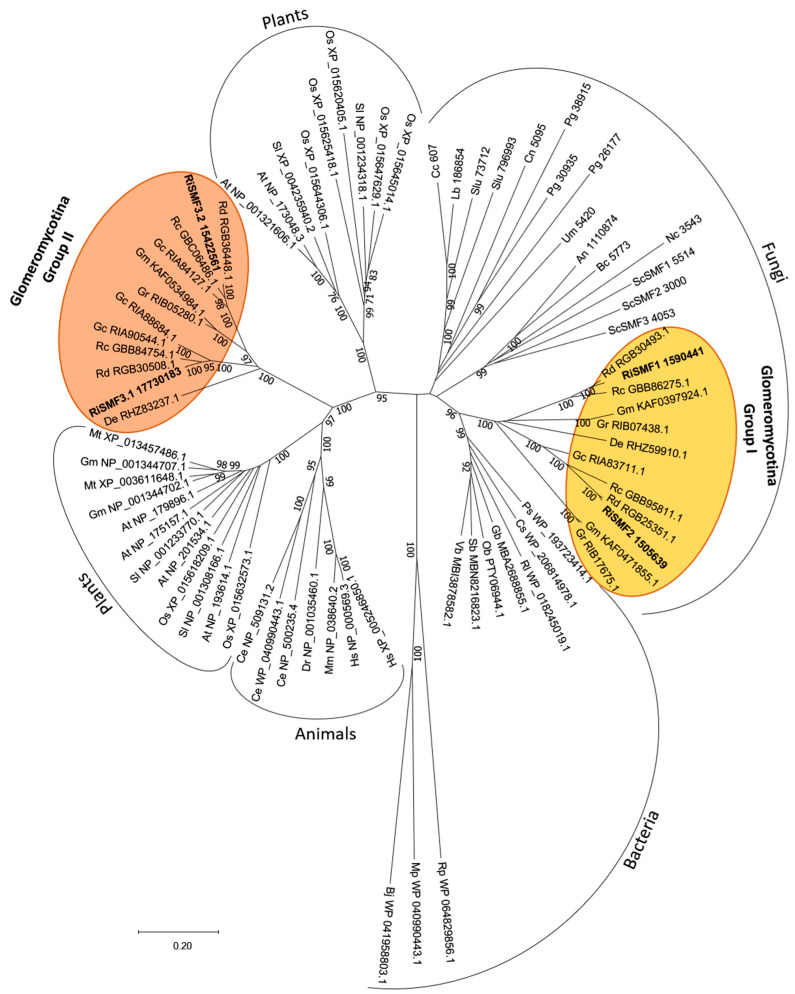
Phylogenetic relationships of the *R. irregularis* NRAMP proteins, with homologous sequences from selected species representative of the different kingdoms. The yellow and orange circles indicate the Glomeromycotan Group I and Group II sequences, respectively. The neighbour-joining tree was created with MEGA-X. Protein JGI or NCBI identification numbers are indicated. Fungi: An, *Aspergillus niger*; Bc, *Botrytis cinerea*; Cc, *Coprinopsis cinerea*; Cn, *Cryptococcus neoformans*; De, *Diversispora epigaea*; Gc, *Glomus cerebriforme*; Gm, *Gigaspora margarita*; Gr, *Gigaspora rosea*; Lb, *Laccaria bicolor*; Nc, *Neurospora crassa*; Pi, *Piriformospora indica*; Pg, *Puccinia graminis*; Rc, *Rhizophagus clarus*; Rd, *Rhizophagus diaphanus*; Ri, *Rhizophagus irregularis*; Sc, *Saccharomyces cerevisiae*; Slu, *Suillus luteus*; Tb, *Tuber melanosporum*; Um, *Ustilago maydis*. Plants: At, *Arabidopsis thaliana*; Gm, *Glycine max*; Mt, *Medicago truncatula*; Os, *Oryza sativa*; Sl, *Solanum lycopersicum*. Bacteria: Bj, *Bradyrhizobium japonicum*; Cs, *Chroococcus sp*; Gb, *Gemmatimonadaceae bacterium*; Mp, *Mesorhizobium plurifarium*; Ob, *Opitutaceae bacterium*; Ps, *Paenibacillus sp*; Rl, *Rhizobium leguminosarum*; Rp, *Rhizobium phaseoli*; Sb, *Spirochaetes bacterium*. Animals: Ce, *Caenorhabditis elegans*; Dr, *Danio rerio*; Hs, *Homo sapiens*; Mm, *Mus musculus*. The *R. irregularis* sequences are in bold.

**Figure 3 jof-08-00592-f003:**
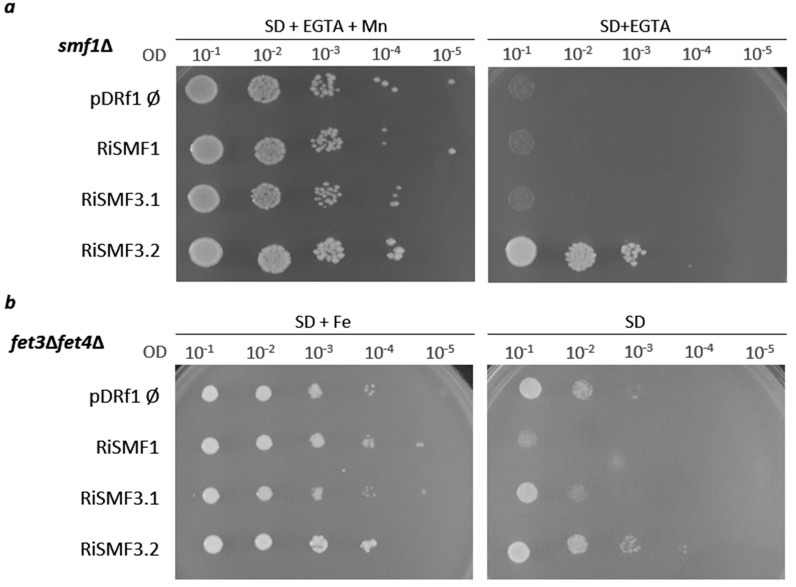
Analysis of RiSMFs function in yeast. (**a**) *smf1*Δ yeast cells transformed with the empty vector or expressing RiSMF1, RiSMF3.1 or RiSMF3.2 were plated on SD-URA medium with 50 mM MES buffer pH 6, 6.25 mM EGTA and supplemented or not with 500 μM MnSO_4_. (**b**) *fet3*Δ*fet4*Δ cells transformed with the empty vector or expressing RiSMF1, RiSMF3.1 or RiSMF3.2 were plated on SD-URA medium pH 3.5, supplemented with 100 µM FeCl_3_, or pH 5.4, non-supplemented with Fe. The cultures were diluted to ODs of 10^−1^ to 10^−5^ (as indicated) and spotted on the corresponding selective media.

**Figure 4 jof-08-00592-f004:**
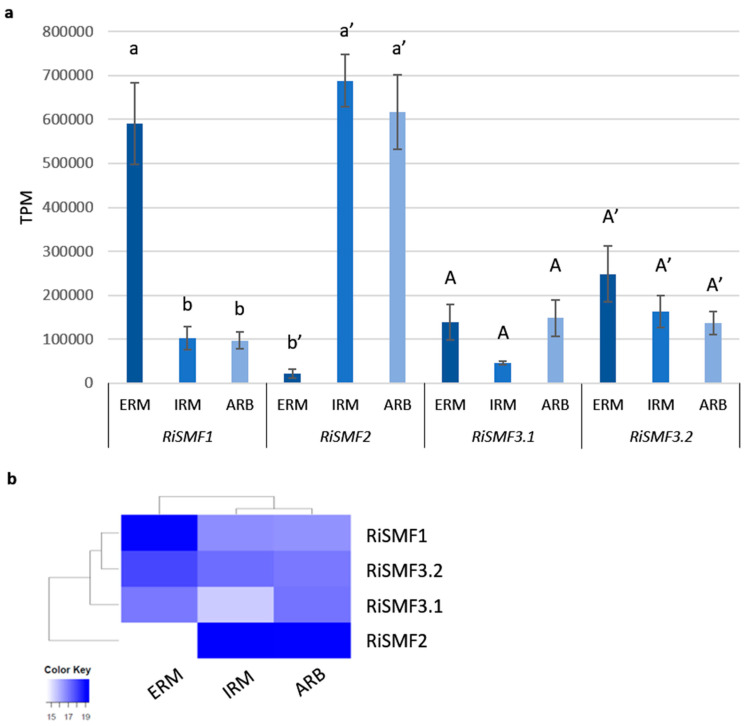
Expression patterns of *R. irregularis SMF* genes. (**a**) *RiSMF*s expression, based on RNAseq analyses of ERM grown in monoxenic cultures as well as IRM and arbucules (ARB) collected by laser microdissection, from *Medicago truncatula* mycorrhizal roots. Error bars represent SE from three biological replicates. Different letters indicate significant differences (*p* < 0.05). (**b**) Heatmap showing the hierarchical clustering of SMF genes, grouped according to the expression patterns in ERM, IRM and ARB. Gradient color ranging from white to bright blue corresponds to expression values log2 transformed.

**Figure 5 jof-08-00592-f005:**
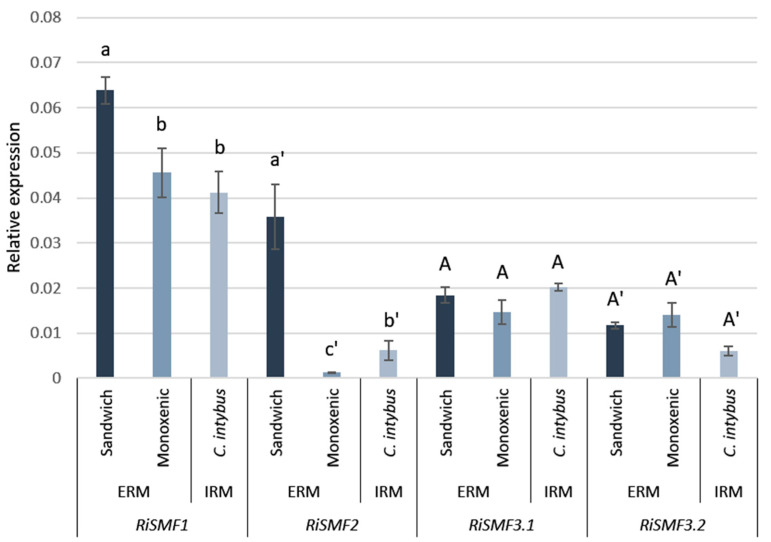
Expression patterns of *R. irregularis SMF* genes. Gene expression was analysed in RNAs isolated from chicory mycorrhizal roots (IRM), ERM collected in the in vivo whole-plant bidimensional culture system and ERM from monoxenic cultures. Relative gene-expression levels were calculated using the 2^−ΔCT^ method, with *RiEF1α* as internal control. Bars represent standard error. Different letter types indicate statistical data for each gene. Different letters indicate significant differences (*p* < 0.05; *n* = 3) between treatments.

**Figure 6 jof-08-00592-f006:**
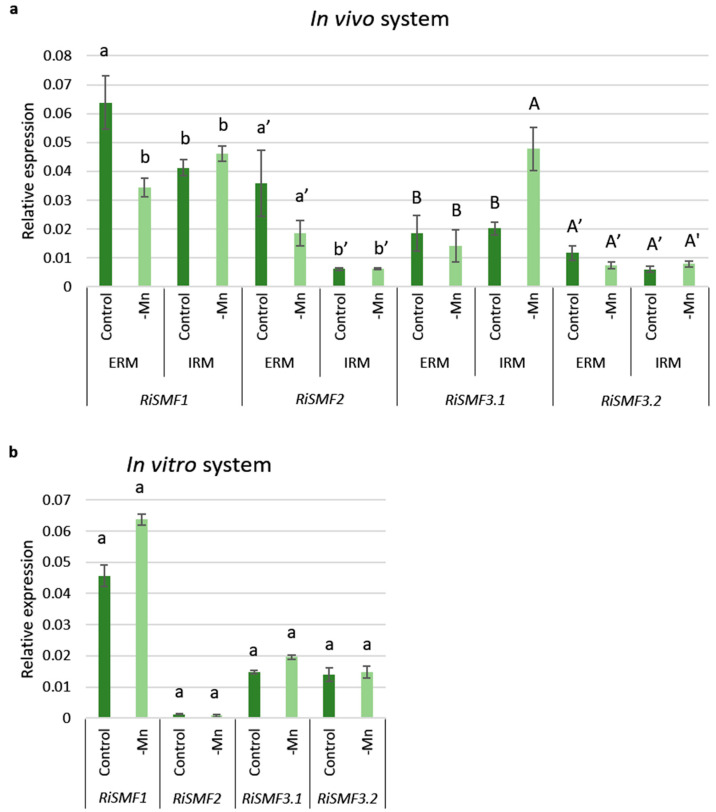
Effect of Mn deficiency on *RiSMF*s expression. (**a**) Gene expression was analysed in RNAs isolated from mycorrhizal roots (IRM) and ERM collected from chicory plants, developed in the whole-plant bidimensional culture system (in vivo system) and watered with half-strength Hoagland solution (control, 4.5 μM) or a modified nutrient solution without Mn (-Mn). (**b**) Gene expression was analysed in RNAs isolated from ERM developed in monoxenic cultures (in vitro system) in liquid M media without Mn (-Mn) or containing 2 μM Mn (control). Relative-gene-expression levels were calculated, using the 2^−ΔCT^ method with *RiEF1α* as internal control. Bars represent standard error. Different letter types indicate statistical data for each gene. Different letters indicate significant differences (*p* < 0.05; *n* = 3) between treatments.

**Figure 7 jof-08-00592-f007:**
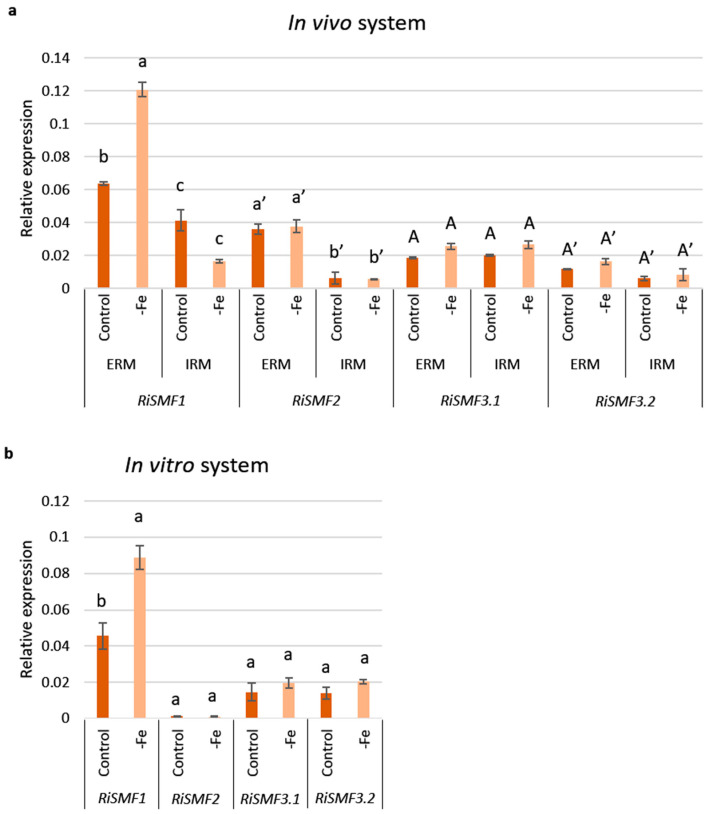
Effect of Fe deficiency on *RiSMF*s expression. (**a**) Gene expression was analysed in RNAs isolated from mycorrhizal roots (IRM) and ERM collected from chicory plants developed in the whole plant bidimensional culture system (in vivo system) watered with half-strength Hoagland solution (control, 50 μM Fe) or a modified nutrient solution without Fe (-Fe). (**b**) Gene expression was analysed in RNAs isolated from ERM developed in monoxenic cultures (*in vitro* system) in liquid M media lacking Fe and exposed 3 days to 0.5 mM ferrozine (-Fe) or in M medium containing 45 μM Fe (control). Relative gene expression levels were calculated using the 2^−ΔCT^ method with *RiEF1α* as internal control. Bars represent standard error. Different type letters indicate statistical data for each gene. Different letters indicate significant differences (*p* < 0.05; *n* = 3) between treatments.

**Figure 8 jof-08-00592-f008:**
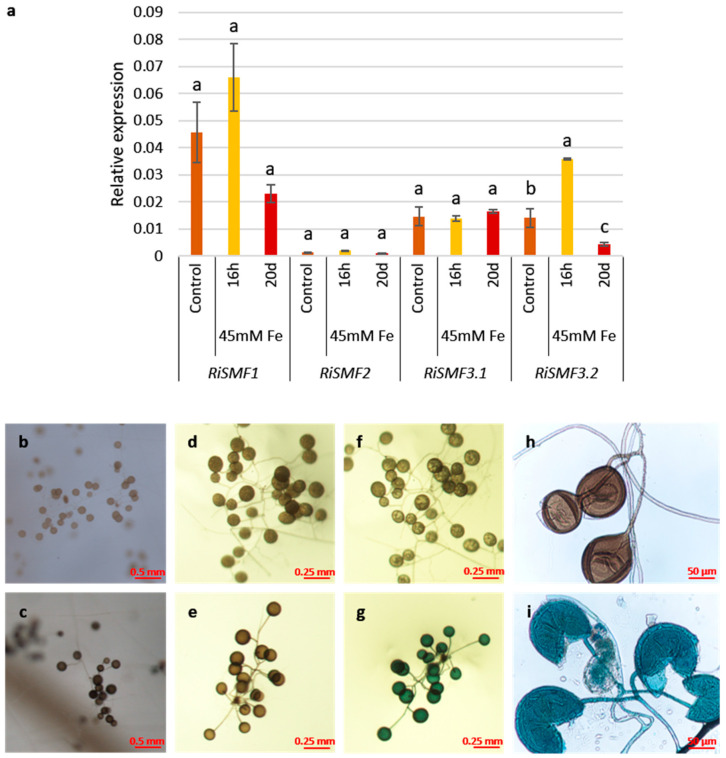
Effect of Fe toxicity on ERM *RiSMF*s expression and Fe accumulation. (**a**) Gene expression was analysed in RNAs isolated from ERM developed in monoxenic cultures in M medium (control), M medium and exposed 3 d for 16 h to 45 mM Fe (16 h) or grown for 20 d in media supplemented with 45 mM Fe (20 d). Relative expression levels were calculated, using the 2^−ΔCT^ method with *RiEF1α* as internal control. Bars represent standard error. Different letters indicate significant differences (*p* < 0.05; *n* = 3), in comparison to the corresponding control value. (**b**–**i**) Fe detection in the ERM by Perls staining. ERM grown in M media supplemented with H_2_O (b, control) or supplemented with 45 mM Fe (c, Fe toxicity). Spores from the control plates (**b**,**d**,**f**,**h**) and from the Fe-toxicity conditions (**c**,**e**,**g**,**i**), before (**d**,**e**) and after (**f**–**i**) the addition of Perls stain solution. Crushed spores developed under control, (**h**) Fe toxicity (**i**) and exposed to Perls stain solution. Blue precipitate indicates Fe accumulation.

**Figure 9 jof-08-00592-f009:**
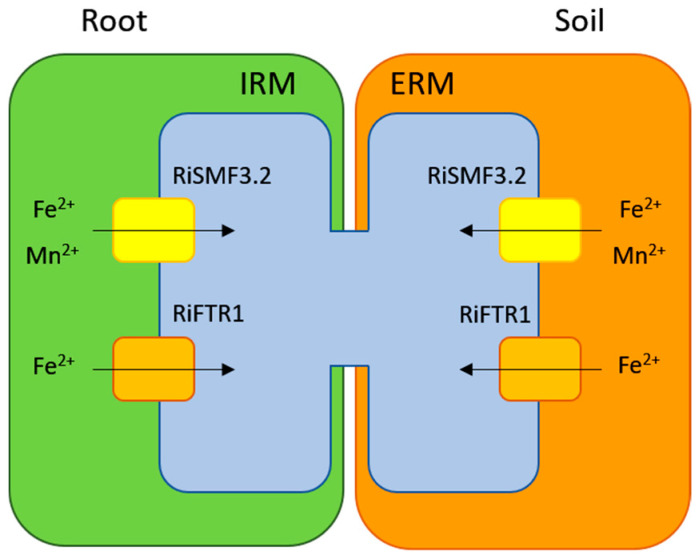
Schematic diagram of the Mn and Fe uptake systems in the ERM and IRM of the AM fungus *R. irregularis*.

**Table 1 jof-08-00592-t001:** Effect of Mn deficiency on ERM development and mycorrhizal colonization.

	Control	-Mn
Spores/cm²	98.67 ± 21.75a	96.28 ± 24.86a
Hyphal length (cm)	93.76 ± 14.07b	197.97 ± 14.22a
RiEF1α	0.20 ± 0.03a	0.15 ± 0.02a
F%	98.60 ± 0.50a	93.09 ± 1.54a
M%	61.93 ± 2.22a	64.94 ± 2.47a
m%	62.71 ± 2.21a	64.62 ± 2.91a
A%	46.61 ± 1.59a	45.77 ± 3.98a
a%	74.98 ± 0.63a	74.66 ± 2.7a

Hyphal length and number of spores were determined in the ERM grown on the nitrocellulose membranes of the whole plant bidimensional culture systems watered with half-strength Hoagland solution (control, 4.5 μM Mn) or with a modified nutrient solution without Mn (-Mn). Mycorrhizal colonization was assessed molecularly by determining the relative expression of *RiEF1α* and histochemically by using the Trouvelot method in roots of chicory roots collected from the culture systems fertilised with and without Mn. F%, frequency of mycorrhiza in the root system; M%, intensity of the mycorrhizal colonization in the root system; m%, intensity of the mycorrhizal colonization in the root fragments; A%, arbuscule abundance in the root system; a%, arbuscule abundance in mycorrhizal parts of root fragments. Values are means ± standard error. Different letters indicate statistically significant differences (*p* < 0.05) among treatments.

**Table 2 jof-08-00592-t002:** Effect of Fe deficiency on ERM development and mycorrhizal colonization.

	Control	-Fe
Spores/cm²	100.21 ± 20.89a	42.50 ± 10.78b
Hyphal length (cm)	95.80 ± 6.01a	115.01 ± 6.12a
RiEF1α	0.18 ± 0.05a	0.20 ± 0.01a
F%	97.12 ± 2.00a	90.88 ± 2.69a
M%	61.93 ± 4.34a	49.55 ± 8.42a
m%	60.75 ± 3.50a	54.27 ± 8.70a
A%	39.36 ± 6.81a	32.04 ± 7.05a
a%	71.95 ± 7.31a	59.72 ± 9.36a

Hyphal length and number of spores were determined in the ERM grown on the nitrocellulose membranes of the whole-plant bidimensional culture systems watered with half-strength Hoagland solution (control, 50 μM Fe) or with a modified nutrient solution without Fe (-Fe). Mycorrhizal colonisation was assessed molecularly, by determining the relative expression of *RiEF1**α*, and histochemically, by using the Trouvelot method in chicory roots collected from the culture systems fertilised with and without Fe. F%, frequency of mycorrhiza in the root system; M%, intensity of the mycorrhizal colonization in the root system; m%, intensity of the mycorrhizal colonization in the root fragments; A%, arbuscule abundance in the root system; a%, arbuscule abundance in mycorrhizal parts of root fragments. Values are means ± standard error. Different letters indicate statistically significant differences (*p* < 0.05) among treatments.

## Data Availability

Not applicable.

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
