# Peer review of "Characterization of the NRAMP Gene Family in the Arbuscular Mycorrhizal Fungus *Rhizophagus irregularis"

_jof, 2022, doi:10.3390/jof8060592_

Round 1
Reviewer 1 Report
Major comments to authors:
The manuscript by López-Lorca and colleagues reports the characterization of the four RiSMF genes including RiSMF1, RiSMF2, RiSMF3.1 and RiSMF3.2 in NRAMP family members of R. irregularis by bioinformatic and phylogenetic analyses, RT-qPCR and heterologous expression in yeasts. However, there are contradictions among the results of protein structures, yeast complementation and subcellular localization presented in this manuscript. Moreover, the writing of this manuscript is not clear in many parts. Thus, before acceptance, these aspects need attention and revision as detailed below to substantiate the claims made.
Detail comments:
- Line 42: what is the mean of “DPGN metal binding domain”? please explain the DPGN?
- Improve the introduction section: many places are not clearly provided. Please focus on the introduction of NRAMP transporters in both fungi and mycorrhizal plants.
- Lines 73-75: it is not clear! Authors should introduce and state these iron and copper transporters’ roles during AM symbiosis and/or AM fungi.
- Lines 76-79: authors should use the latest genomic (Morin et al., 2019) and transcriptomic (Zeng et al., 2018, 2020) data for irregularis DAOM197198 to identify all members of NRAMP family.
- Lines 46-47: please explain SMF1-3?
- Lines 82-84: also need to use new data (Morin et al., 2019; Zeng et al., 2018, 2020) to search all members of RiSMFs or use the hmm model to scan the whole genome.
- Lines 90-91: authors should use the SWISS model to construct 3D-structures of these NRAMP proteins.
- Line 95: for MEGA-X, need to cite one reference here.
- Lines 98-99: please simplify the AMF name as Rhizophagus irregularis
- Line 128: how many AMF spores were contained in the 3 ml of AM fungal inoculum?
- Please combine 2.3 (lines 149-156) and 2.6 (lines 189-204) sections into one.
- Lines 164-165: do the authors experimentally validate the existence of this RiSMF2 in genome of irregularis DAOM197198? Please clone and provide the genomic sequence of RiSMF2 based on PCR method.
- Lines 171-172: 5 μL of serial 1:10 dilutions were spotted on the corresponding selective medium. However, the results of yeast growth (Figure 3 and Figure S1a) were not shown as the serial 1:10 dilutions. Thus, in Section 3.2, the concentration of yeast cells should be diluted into cell suspension with different concentrations in functional characterization of RiSMFs in yeast strains.
- Lines 172-173: pH3.5, pH5.4? Why? Please explain this difference for pH values? since yeast cell growth is obviously affected under very low pH conditions.
- Lines 177-188: in RiSMFs subcellular localization experiment, lacking positive control experiment with well-known SMF protein location as well as PM marker with red fluorescence.
- Section 2.5. and Figure S1b: In the experiment of RiSMFs subcellular localization, the vector pGWFDR196 harboring marker protein (pGWFDR196::eGFP) should be used as control, whereas in Figure S1b only pGWFDR196Ø was shown, please correct it.
- Line 203: why not use 2-ΔΔCt method to calculate the relative expression level?
- Line 207 and Line 210: please explain PERLs reagent and Perls stain solution with detail components?
- Figure 1 and Figure S1b: RiSEMFs contain 11-12 TM domains, indicating that they may be located into PM or endomembrane systems, however, Figure S1b showed that these RiSMFs were located at cytoplasm. From these results, RiSMFs seem to be not correctly localized in yeast cells. Moreover, need to improve the subcellular localization experiment by other approaches.
- Figure 3: complementation of RiSMFs in yeast cells: (1) RiSMF1 and RiSMF3.1 cannot restore the phenotype of the yeast smfΔ cells, due to their wrong location in yeast. Please see Figure S1b, both RiSMF1 and RiSMF3.1 were localized at cytoplasm but not at PM. (2) RiSMF3.2 seems to be partially grown in yeast cells, but it was not localized into PM, whereas the yeast SMF protein is present into PM. Thus, these results shown in Figure 3 also are contradictory.
- Heatmap of the expression profiles of these several NRAMP transporters encoding genes in different structures of irregularis should be generated with the latest transcriptome data (Zeng et al., 2018, 2020).
Zeng, T.; Rodriguez-Moreno, L.; Mansurkhodzaev, A.; Wang, P.; van den Berg, W.; Gasciolli, V.; Cottaz, S.; Fort, S.; Thomma, B.; Bono, J.J.; et al. A lysin motif effector subverts chitin-triggered immunity to facilitate arbuscular mycorrhizal symbiosis. New Phytol 2020, 225, 448-460, doi:10.1111/nph.16245.
Zeng, T.; Holmer, R.; Hontelez, J.; Te Lintel-Hekkert, B.; Marufu, L.; de Zeeuw, T.; Wu, F.; Schijlen, E.; Bisseling, T.; Limpens, E. Host- and stage-dependent secretome of the arbuscular mycorrhizal fungus Rhizophagus irregularis. Plant J 2018, 94, 411-425, doi:10.1111/tpj.13908.
- Figure 4, data analysis: using different letter types to indicate the statistical data in the same group. For example, a/a, a’/a’, or A/A, A/B.
- The titles of tables are too long! Please simplify them and add the details below the tables as notes.
- Section 4: In the Discussion part, after the major revision, the authors should improve the discussion contents and conclusions as well as provide a schematic diagram of the working model of Mn and Fe transporters such as RiSMFs in different structures of R. irregularis, which will show this work more clear.
Author Response
We would like to thank this Reviewer for his/her careful reading of our manuscript and for all his/her constructive comments, which indeed have improved the quality of our manuscript. We have revised the manuscript taking into account the detailed comments. Many parts of the manuscript have been rewritten and we have answered all questions/concerns raised by this Reviewer (in blue). Line numbers refer the ones of the revised version with track changes.
Detail comments:
1. Line 42: what is the mean of “DPGN metal binding domain”? please explain the DPGN?
The DPGN metal binding domain is a motif that is present in most eukaryotes and prokaryotes NRAMP sequences. Numerous mutational studies have shown that this motif is essential for metal transport of the NRAMP transporters (Brozzi and Gaudet. 2021. Molecular mechanisms of Nramp-family transition metal transport. J Mol Biol 443: 166991). For example, either alanine or more conservative substitutions for the aspartate and asparagine (D56 and N59) impaired metal (II) binding to the ScaNramp of Staphylococcus capitis (Ehrnstorfer et al. 2014. Crystal structure of a SLC11(NRAMP) transporter reveals the basis for transition metal ion transport. Nat Struct Mol Biol 21: 990–996) and eliminated or severely reduced metal transport in human NRAMP2 (Bozzi et al. 2016. Conserved methionine dictates substrate preference in Nramp-family divalent metal transporters. Proc Natl Acad Sci USA 113: 0310–10315; Lam-Yuk-Tseung et al. 2003. Iron transport by Nramp2/DMT1: pH regulation of transport by 2 histidines in transmembrane domain 6. Blood 101: 3699–3707) and EcoliNramp (Chaloupka et al. 2005. Identification of functional amino acids in the Nramp family by a combination of evolutionary analysis and biophysical studies of metal and proton cotransport in vivo. Biochemistry, 44: 726–733; Haemig and Brooker. 2004. Importance of conserved acidic residues in mntH, the Nramp homolog of Escherichia coli. J Membr Biol 201: 97–107). Therefore, these two-metal coordinating residues appear essential to the general Nramp transport mechanism, such that no metal substrate will bind without a suitable placed residue in these positions.
The text has been modified to clarify what is the DPGN metal binding domain (lines 42-44).
2. Improve the introduction section: many places are not clearly provided. Please focus on the introduction of NRAMP transporters in both fungi and mycorrhizal plants.
The Introduction section has been extended. We have described the most significant data on NRAMP fungal transporters (lines 57-66) and what is known about plant NRAMP transporters in mycorrhizal roots (Lines 95-103).
3. Lines 73-75: it is not clear! Authors should introduce and state these iron and copper transporters’ roles during AM symbiosis and/or AM fungi.
The roles of the R. irregularis iron and copper transporters have been described in the revised manuscript (Lines 87-93).
4. Lines 76-79: authors should use the latest genomic (Morin et al., 2019) and transcriptomic (Zeng et al., 2018, 2020) data for irregularis DAOM197198 to identify all members of NRAMP family.
Many thanks for this comment. We have mined the R. irregularis genomic data of Morin et al. and the genomic data of different R. irregularis isolates (Chen et al. 2018. High intraspecific genome diversity in the model arbuscular mycorrhizal symbiont Rhizophagus irregularis. New Phytol 220: 1161–1171). In this search we have found the same four genes that we found in the R. irregularis genomic data of Tisserant et al. We have provided the reference of Chen et al. 2018 instead of the reference of Morin et al because these authors indicate that the genome assembly and annotation from R. irregularis DAOM197198 is given in Chen et al. 2018.
We also mined the R. irregularis DAOM197198 transcriptomic data of Zeng et al, and found the same four genes.
In the Materials and Methods section we have indicated that we have used all these databases to identify the members of the NRAMP family of R. irregularis (lines 112-116). Discussion has also been modified to address this point (lines 571-573).
5. Lines 46-47: please explain SMF1-3?
Yeast NRAMP proteins were named as SMFs because they were initially identified as hydrophobic proteins that suppress a lethal mutation in the yeast mitochondrial processing enhancing proteins. Therefore, the name SMF comes from Suppressor of Mitochondrial import Function (West et al. 1992. Two related genes encoding extremely hydrophobic proteins suppress a lethal mutation in the yeast mitochondrial processing enhancing protein. J Biol Chem 267: 24625-24633).
The meaning of SMF has been included in the manuscript (lines 48-50).
6. Lines 82-84: also need to use new data (Morin et al., 2019; Zeng et al., 2018, 2020) to search all members of RiSMFs or use the hmm model to scan the whole genome.
As indicated in point 4, the new transcriptome and genome datasets of R. irregularis have been searched to identify NRAMP sequences. These analyses revealed that the RiSMF family is composed of four members.
7. Lines 90-91: authors should use the SWISS model to construct 3D-structures of these NRAMP proteins.
The 3D-structures of the four R. irregularis NRAMP proteins have been constructed and included in the manuscript as Figure 1d.
8. Line 95: for MEGA-X, need to cite one reference here.
A reference has been included to cite MEGA-X
9. Lines 98-99: please simplify the AMF name as Rhizophagus irregularis
The name of Rhizophagus irregularis has been simplified. However, we have maintained the ecotype used.
10. Line 128: how many AMF spores were contained in the 3 ml of AM fungal inoculum?
The 3 ml of AM fungal inoculum used to inoculate the plants contained 1650 spores. This has been included in line 166.
11. Please combine 2.3 (lines 149-156) and 2.6 (lines 189-204) sections into one.
As it has been suggested, these two sections have been merged into a single one.
12. Lines 164-165: do the authors experimentally validate the existence of this RiSMF2 in genome of irregularis DAOM197198? Please clone and provide the genomic sequence of RiSMF2 based on PCR method.
As indicated in Materials and Methods section in lines 219-220, a PCR band of the expected size was obtained with cDNA or genomic DNA from extraradical mycelium and the RiMSF2-gene specific primers designed to amplify the full-length RiMSF2 gene. However, all attempts we did to clone the RiSMF2 full-length cDNA failed. As it has been previously reported by several authors (Bernaudat et al. 2011. Heterologous expression of membrane proteins: choosing the appropriate host. PLos One 6: e29191; Anthony et al. 2004. Tightly regulated vectors for the cloning and expression of toxic genes. J Microbiol Methods 58: 243-250), cloning of genes encoding membrane proteins in E. coli is often difficult because these proteins, as a consequence of their hydrophobicity, are lethal to E. coli at an extremely low quantity of molecules per cell. We tried different strategies to prevent unwanted expression of the toxic protein during culture propagation, such as using different cloning vectors, insert/vector ratios, E. coli culture conditions and strains. Unfortunately, all our attempts to clone RiSMF2 failed.
The existence of RiMSF2 in the R. irregularis genome was also verified by our gene expression analyses since the RiSMF2 transcripts were detected in all fungal materials used in this study.
13. Lines 171-172: 5 μL of serial 1:10 dilutions were spotted on the corresponding selective medium. However, the results of yeast growth (Figure 3 and Figure S1a) were not shown as the serial 1:10 dilutions. Thus, in Section 3.2, the concentration of yeast cells should be diluted into cell suspension with different concentrations in functional characterization of RiSMFs in yeast strains.
Many thanks for this comment. We apologize for this mistake. Figures 3 and S1a have been modified to illustrate growth of serial 1:10 dilutions.
14. Lines 172-173: pH3.5, pH5.4? Why? Please explain this difference for pH values? since yeast cell growth is obviously affected under very low pH conditions.
The Dfet3Dfet4 double mutant has a severe impairment in Fe uptake, requiring high iron levels in the medium and a more acidic environment to increase this micronutrient mobility and availability. Therefore, the Fe-replete positive controls were obtained by growing the double mutant on pH 3.5 medium supplemented with 100 mM FeCl3. However, the phenotypic tests were performed in SD media (pH 5.4) non-supplemented with Fe. The text has been modified to explain why the Fe-replete positive controls were grown at pH 3.5 and not at pH 5.4 (SD medium). Lines 377-383.
14. Lines 177-188: in RiSMFs subcellular localization experiment, lacking positive control experiment with well-known SMF protein location as well as PM marker with red fluorescence.
We absolutely agree with the Reviewer that we should have included a positive control and a fluorescent plasma membrane marker in the subcellular localization experiment. However, to us the positive controls are not essential for the interpretation of our data. Therefore, we would like to ask this Reviewer to allow us not to provide the positive control data. Obtaining the positive controls might be time-consuming, as we have to make the required construct and to perform the phenotypic and localization experiments.
From our point of view, our data clearly illustrate that only RiSMF3.2 is functional in the heterologous system (Fig. 3) and that the RiSMF3.2-GFP fusion protein is not functional (Fig. S1). Although subcellular localization of RiSMF3.2 could not be experimentally demonstrated in yeast, the finding that it complements the growth defect of a mutant yeast strain lacking the plasma membrane transporter SMF1 indicates that RiSMF3.2 is targeted to the plasma membrane.
Our localization assays show that the RiSMF-GFP proteins are not targeted to any yeast membrane. Although the principles of targeting seem to be conserved between organisms, the heterologous proteins may lack the sequences required for targeting to the correct compartment in the cell and mistargeting can occur. Our data suggest that the RiSMF-GFP recombinant proteins are not properly folded and that they are likely degraded in the cytoplasmic proteasomes. As we indicated in the manuscript, this is a common problem observed in yeast membrane proteins expression assays.
15. Section 2.5. and Figure S1b: In the experiment of RiSMFs subcellular localization, the vector pGWFDR196 harboring marker protein (pGWFDR196::eGFP) should be used as control, whereas in Figure S1b only pGWFDR196Ø was shown, please correct it.
We are very sorry for this mistake. It has been corrected.
16. Line 203: why not use 2-ΔΔCt method to calculate the relative expression level?
The 2-DCT method is a variation of the Livak (2-DDCT) method that is simpler to perform and gives essentially the same results. The 2-DCT method also uses the difference between reference and target CT values for each sample. The key difference in the results is that the expression value of the calibrator sample is not 1.0. If the resulting expression values obtained with the 2-DCT method are divided by the expression value of a chosen calibrator, the results are exactly the same as those obtained with the 2-DDCT method. To us the 2-DCT method gives more information because shows expression level of the target gene in each sample or experimental condition. The 2-DDCT method gives the fold induction without knowing the levels of expression. Therefore, when analysing the expression of different genes, the 2-DCT method is more informative because allows to compare the relative expression of different genes in each sample.
17. Line 207 and Line 210: please explain PERLs reagent and Perls stain solution with detail components?
The principle of Perl´s Prussian blue reaction is that potassium ferrocyanide will form ferric ferrocyanide (Prussian blue) with reactive ferric salts in an acid solution. The addition of hydrochloric acid increases the availability of iron within the tissue for reaction with the potassium ferrocyanide.
The detailed components of the Perls stain solution are indicated in Materials and Methods (section 2.7). The principle of this reaction has been explained in Results section (lines 492-495).
18. Figure 1 and Figure S1b: RiSEMFs contain 11-12 TM domains, indicating that they may be located into PM or endomembrane systems, however, Figure S1b showed that these RiSMFs were located at cytoplasm. From these results, RiSMFs seem to be not correctly localized in yeast cells. Moreover, need to improve the subcellular localization experiment by other approaches.
We agree that in the heterologous system the RiSMFs were located at the cytoplasm of the yeast cells. To us this is an artefact of the heterologous system and not because there is a contradiction between our in silico and yeast complementation analyses. As indicated in point 15, problems regarding the correct folding and targeting of the heterologous expressed protein can occur (Frommer and Ninnemann. 1995. Heterologous expression of genes in bacterial, fungal, animal and plant cells. Annu Rev Plant Physiol Plan Mol Biol 46: 419-444). We tried to improve the subcellular localization experiment by growing the yeast cells in different media and temperatures. However, all these strategies failed. As indicated in the manuscript, detection of the RiSMF-GFP proteins in the cytoplasm is indicative of proteolysis in the cytosolic proteasomes. Degradation of the RiSMF proteins in the yeast cells could be expected since the S. cerevisiae SMF1 and SMF2 transporters are regulated at the post-translational level. In the presence of high metal levels they are degraded to avoid excessive metal accumulation in the cytosol. If high amounts of the RiSMF proteins are expressed in the yeast cells, they might be degraded to avoid excessive import of metals in the RiSMF-expressing cells.
19. Figure 3: complementation of RiSMFs in yeast cells: (1) RiSMF1 and RiSMF3.1 cannot restore the phenotype of the yeast smfΔ cells, due to their wrong location in yeast. Please see Figure S1b, both RiSMF1 and RiSMF3.1 were localized at cytoplasm but not at PM. (2) RiSMF3.2 seems to be partially grown in yeast cells, but it was not localized into PM, whereas the yeast SMF protein is present into PM. Thus, these results shown in Figure 3 also are contradictory.
As indicated in point 19, to us localization of the RiSMF1-GFP, RiSMF3.-GFP and RiSMF3.2–GFP fusion proteins at the cytoplasm is due to an artefact of the heterologous system. As indicated in point 15, we believe that addition of the GFP tag to RiSMF3.2 has produced a change in its structure preventing proper folding and targeting to the yeast plasma membrane. Actually, while the RiSMF3.2 protein was functional in yeast, the RiSMF3.2-GFP fusion protein was not functional. Therefore, to us there is not a contradiction between data of Figures S1 and 3.
20. Heatmap of the expression profiles of these several NRAMP transporters encoding genes in different structures of irregularis should be generated with the latest transcriptome data (Zeng et al., 2018, 2020).
Zeng, T.; Rodriguez-Moreno, L.; Mansurkhodzaev, A.; Wang, P.; van den Berg, W.; Gasciolli, V.; Cottaz, S.; Fort, S.; Thomma, B.; Bono, J.J.; et al. A lysin motif effector subverts chitin-triggered immunity to facilitate arbuscular mycorrhizal symbiosis. New Phytol 2020, 225, 448-460, doi:10.1111/nph.16245.
Zeng, T.; Holmer, R.; Hontelez, J.; Te Lintel-Hekkert, B.; Marufu, L.; de Zeeuw, T.; Wu, F.; Schijlen, E.; Bisseling, T.; Limpens, E. Host- and stage-dependent secretome of the arbuscular mycorrhizal fungus Rhizophagus irregularis. Plant J 2018, 94, 411-425, doi:10.1111/tpj.13908.
Many thanks for this suggestion. We have mined the Zeng et al transcriptimic data and provided a heatmap of the expression profiles of the R. irregularis NRAMP family members in Figure 4. A new section has been included to describe these results.
21. Figure 4, data analysis: using different letter types to indicate the statistical data in the same group. For example, a/a, a’/a’, or A/A, A/B.
Many thanks for this suggestion. Figures 4 and 5 (5 and 6 in the revised version) have been modified and different letter types have been used to indicate the statistical data for each gene.
22. The titles of tables are too long! Please simplify them and add the details below the tables as notes.
As suggested the titles of tables have been shortened. The additional information has been provided as footnotes.
23. Section 4: In the Discussion part, after the major revision, the authors should improve the discussion contents and conclusions as well as provide a schematic diagram of the working model of Mn and Fe transporters such as RiSMFs in different structures of R. irregularis, which will show this work more clear.
The discussion section has been revised and the contents have been improved. A schematic diagram of the working model of Mn and Fe transporters identified so far in the arbuscules and extraradical mycelium is provided in Figure 7.

Reviewer 2 Report
Dear authors, I have read with interest the manuscript entitled "Characterization of the NRAMP gene family in the arbuscular mycorrhizal fungus Rhizophagus irregularis".
The entire manuscript presents well the topic and the results and discussions are well written. It is interesting that you provide a novel report on a species, which is important to be replicated in future studies.
There is a minor suggestion that I consider will improve the manuscript.
Make a separate paragraph at the end of the Introduction section to present only the aim of your research and the hypotheses or possible outcomes. This will make a reader more curious about what will be presented in the next parts of the manuscript.
Author Response
We are very grateful to this Reviewer for his/her careful reading of our manuscript and for the constructive and very positive comments.
There is a minor suggestion that I consider will improve the manuscript.
Make a separate paragraph at the end of the Introduction section to present only the aim of your research and the hypotheses or possible outcomes. This will make a reader more curious about what will be presented in the next parts of the manuscript.
Many thanks for this comment, which indeed will increase the interest of the readers. As suggested, the aim of our work is presented in a separated paragraph and we have added a new sentence to indicate the possible outcomes.
Round 2
Reviewer 1 Report
Manuscript ID: jof-1696479-v2
Major comments to authors:
The 2nd round manuscript by López-Lorca and colleagues reports the characterization of the four RiSMF genes including RiSMF1, RiSMF2, RiSMF3.1 and RiSMF3.2 in NRAMP family members of R. irregularis by bioinformatic analysis, RT-qPCR as well as heterologous expression in yeasts. This study is interesting. They partially improved the MS according to the comments, however, there are still contradictions between the results of yeast complementation and subcellular localization presented, and also between RNA-seq data (Heatmap) and RT-qPCR data (Figures 5-7) in this updated manuscript. Moreover, some responses did not meet my questions and the authors could not take some times to revise the major issues as mentioned during 1st review. Thus, I cannot accept the current version of the presentation, but I can give the authors another chance and more time to improve them as detailed below to substantiate the claims made.
Detail comments:
- Figure 1d, for 3D structures of NRAMP transporters, lacking template’s protein code or model structure, please provide them and shown it in the figure legend.
- Figure 3, RiSMF3.2 can rescue the yeast smf1 mutant, but Fe/Mn contents of yeast cells should be tested. Moreover, RiSMF3.2 was not localized into PM of yeast. How to complement the corresponding smf1 mutant (a PM protein mutant)???
- What is a pity! RiSMF2 was not isolated. Based on RNA-seq data (Figure 4), RiSMF2 is very more important than other RiSMFs, and is specially expressed in IRM including arbuscules, indicative of its role in re-uptake excess heavy metals (HMs) from PAS or apoplast to AMF in order to reduce HMs in plants. Thus, the authors should take some time to characterize this RiSMF2
- Lines 171-172: 5 μL of serial 1:10 dilutions were spotted on the corresponding selective medium. However, the results of yeast growth (Figure 3 and Figure S1a) were not shown as the serial 1:10 dilutions. Thus, in Section 3.2, the concentration of yeast cells should be diluted into cell suspension with different concentrations in functional characterization of RiSMFs in yeast strains.
- Figure 4 need to be improved. Moreover, this heatmap shown was not clear, without TPM values (see Zeng et al., 2020, NPH, Fig.1a and 1c).
- Figure S1b showed that these RiSMFs were located at cytoplasm. From these results, RiSMFs seem to be not correctly localized in yeast cells. Moreover, need to improve the subcellular localization experiment by other approaches, since authors cannot finish this in yeast cells. They can generate Pirospora indica protoplasts to localize these RiSMFs, or conduct the immunolocalization experiment in AM fungus as their own previous study (Pérez-Tienda et al., 2011, Fungal Genetics and Biology, 48(11):1044-1055).
- RiSMF2 expression patterns are not correct: in RNA-seq analysis (Figure 4), RiSMF2 expression is much higher in IRM than in ERM (almost undetectable in ERM), but RiSMF2 expression is much higher in ERM than in IRM in RT-qPCR experiments (Figures 5a and 6a). Second, in RNA-seq analysis, RiSMF2 expression is much higher than expression of RiSMF1 and 3.1/3.2 in IRM, but this pattern is opposite in RT-qPCR experiments (Figures 5-7)
- In the Discussion part (Figure 8), after the revision, the authors provided a schematic diagram of the working model of Mn and Fe transporters such as RiSMFs in different structures of R. irregularis, but it is not conclusive and not clear! First, these SMFs are not demonstrated to localize to PM. Second, RiSMF2 is mainly expressed in IRM, but not shown in this model.
Author Response
The 2nd round manuscript by López-Lorca and colleagues reports the characterization of the four RiSMF genes including RiSMF1, RiSMF2, RiSMF3.1 and RiSMF3.2 in NRAMP family members of R. irregularis by bioinformatic analysis, RT-qPCR as well as heterologous expression in yeasts. This study is interesting. They partially improved the MS according to the comments, however, there are still contradictions between the results of yeast complementation and subcellular localization presented, and also between RNA-seq data (Heatmap) and RT-qPCR data (Figures 5-7) in this updated manuscript. Moreover, some responses did not meet my questions and the authors could not take some times to revise the major issues as mentioned during 1st review. Thus, I cannot accept the current version of the presentation, but I can give the authors another chance and more time to improve them as detailed below to substantiate the claims made.
We would like to thank this Reviewer for his detailed reading of the revised version of our manuscript and for the time he devoted to it. We did our best to answer all the points raised. We are very sorry if our answers did not meet his expectations and apologize if we did not explain ourselves properly.
As we tried to explain in our rebuttal and as it is explained below, we do not see any contradiction between results of the yeast complementation and subcellular localization assays. We hope this time we have explained better our point of view and that the Reviewer understand our rational.
Concerning the issue of the RNAseq and RT-qPCR data, the updated Fig. 5 shows that the expression levels of RiSMF2 is different in ERM grown in monoxenic cultures and in the in vivo whole plant experimental system, which might be due to the different physiological and developmental stages of the ERM grown under these two experimental systems. Roots used in transformed root organ cultures, compared with those of whole plants, have an altered hormonal balance and a different way of acquiring carbohydrates, which may affect multiple physiological traits of the symbiotic interaction.
If we compare RiSMF2 expression levels in the IRM of our chicory roots with those of our ERM developed in monoxenic cultures, RiSMF2 transcript levels are higher in the IRM. These data are in agreement with those of the RNAseq analyses, that also used ERM from monoxenic cultures. In the former version, we compared RiSMF2 expression in the IRM and ERM developed in the in vivo whole plant sandwich system.
However, there is still a discrepancy between the expression levels of RiSMF1 and RiSMF2 in the IRM obtained in the RNAseq and RT-qPCR analyses, which might be because expression of these genes in the IRM depend on the host genotype or on the experimental conditions. Host-dependent expression of a subset of R. irregularis secreted proteins was also reported by Zheng et al., 2018.
Detail comments:
1. Figure 1d, for 3D structures of NRAMP transporters, lacking template’s protein code or model structure, please provide them and shown it in the figure legend.
We apologize for not including the template´s protein code or model structure. The code has been included in Materials and Methods and in legend of Figure 1D. The model structure of Eremoccus coleosa MntH has been included in Figure 1D.
2. Figure 3, RiSMF3.2 can rescue the yeast smf1 mutant, but Fe/Mn contents of yeast cells should be tested. Moreover, RiSMF3.2 was not localized into PM of yeast. How to complement the corresponding smf1 mutant (a PM protein mutant)???
The protein that was not localized into yeast plasma membrane was RiSMF3.2-GFP. Localization of RiSMF3.2 could not be experimentally determined, but the complementation assays indicate that it is located at the yeast plasma membrane.
RiSMF3.2 recued the yeast smf1 and fet3fet4 mutants (Fig. 3), indicating that RiSMF3.2 has both a Mn and Fe transport function. However, the RiSMF3.2-GFP fusion protein did not rescue the phenotype of any of the two strains (Fig. S1), which means that adding the GFP tag to RiSMF3.2 inactivated its transport function. Therefore, our localization assays just reveal the RiSMF3.2-GFP fusion protein did not have any transport function because it was not targeted to the plasma membrane. As we indicated in the manuscript and in our rebuttal, this is a frequent phenomenon when expressing recombinant membrane proteins tagged with GFP in yeast. This artefact hampered determining subcellular location of RiSMF3.2.
The finding that RiSMF3.2 rescued the phenotype of the yeast strains lacking the plasma membrane transporters smf1 indicates that RiSMF3.2 is the orthologue of yeast smf1 and that RiSMF3.2 is located in the plasma membrane. Therefore, to us there is no contradiction between the complementation and localization assays. It is just that we could not determine experimentally subcellular location of RiSMF3.2 because this protein lost its function when the GFP tag was added to the coding sequence.
Fe/Mn contents of yeast cells could be provided and indeed would support our phenotypic data. However, it is impossible to provide these data in the time given to us to resubmit our manuscript (5 days). Since these data are not essential to define RiSMF3.2 function, we will be very grateful if this Reviewer allows us not to provide them. If he still believes we should provide the Mn/Fe content of the yeast cells, we would like to ask for an extended deadline (2-3 weeks).
3. What is a pity! RiSMF2 was not isolated. Based on RNA-seq data (Figure 4), RiSMF2 is very more important than other RiSMFs, and is specially expressed in IRM including arbuscules, indicative of its role in re-uptake excess heavy metals (HMs) from PAS or apoplast to AMF in order to reduce HMs in plants. Thus, the authors should take some time to characterize this RiSMF2
We agree that the RNAseq analyses indicate that RiSMF2 might be the most important isoform in the IRM in Medicago roots. We agree that an in-depth study of this isoform should be performed. However, we have been working on the isolation of RiSMF2 for more than 3 years. As indicated in our rebuttal, we used different cloning strategies and tried to optimize the cloning protocols, but all our attempts to clone RiSMF2 failed. We obtained a PCR band of the expected size, but all the colonies we obtained had the empty vector. Since we could not afford spending any more time and money in this issue and since we found all the other data interesting, we decided to prepare a manuscript with the data we had. We really hope the Reviewer understand that some genes encoding highly hydrophobic proteins are very difficult to clone. In fact, we talked to Prof. Sebastian Thomine, an authority on plant NRAMP transporters, and confirmed us that some NRAMP proteins cannot be cloned.
4. Lines 171-172: 5 μL of serial 1:10 dilutions were spotted on the corresponding selective medium. However, the results of yeast growth (Figure 3 and Figure S1a) were not shown as the serial 1:10 dilutions. Thus, in Section 3.2, the concentration of yeast cells should be diluted into cell suspension with different concentrations in functional characterization of RiSMFs in yeast strains.
Updated Figures 3 and S1a show serial 1:10 dilutions. Yeast cells were grown to a final OD of 1. Then, 5 microL of serial 1:10 dilutions were spotted on the corresponding selective medium. We have included the ODs in the Figure and modified the figure legends to indicate that serial 1:10 dilutions were spotted.
5. Figure 4 need to be improved. Moreover, this heatmap shown was not clear, without TPM values (see Zeng et al., 2020, NPH, Fig.1a and 1c).
Figure 4 has been improved. We have provided an additional figure illustrating TPM values.
6. Figure S1b showed that these RiSMFs were located at cytoplasm. From these results, RiSMFs seem to be not correctly localized in yeast cells. Moreover, need to improve the subcellular localization experiment by other approaches, since authors cannot finish this in yeast cells. They can generate Pirospora indica protoplasts to localize these RiSMFs, or conduct the immunolocalization experiment in AM fungus as their own previous study (Pérez-Tienda et al., 2011, Fungal Genetics and Biology, 48(11):1044-1055).
We agree that we cannot finish the subcellular experiment in yeast and that we should use a different experimental approach. Many thanks for the suggestion of expressing the RiSMF-GFP proteins in Piriformospora indica or to raise specific antibodies against the different RiSMF proteins for immunolocalization. We think both ideas are great, but both experiments are highly time-consuming and will be the objective of further studies. As this Reviewer knows, it is not possible to provide these results during the revision process of this manuscript.
7. RiSMF2 expression patterns are not correct: in RNA-seq analysis (Figure 4), RiSMF2 expression is much higher in IRM than in ERM (almost undetectable in ERM), but RiSMF2 expression is much higher in ERM than in IRM in RT-qPCR experiments (Figures 5a and 6a). Second, in RNA-seq analysis, RiSMF2 expression is much higher than expression of RiSMF1 and 3.1/3.2 in IRM, but this pattern is opposite in RT-qPCR experiments (Figures 5-7).
To address this point we have included an additional figure to compare expression levels of the RiSMFs in IRM and ERM grown in the in vivo sandwich system and in the ERM grown in the monoxenic cultures. As it is shown in Fig. 5, expression level of RiSMF2 is much lower in the ERM grown in monoxenic cultures than in the in vivo sandwich system. These data indicate that ERM RiSMF2 expression in the ERM is modulated by the developmental and physiological stage of the fungus.
If we compare RiSMF2 expression in IRM (from the in vivo whole plant sandwich) and ERM from monoxenic cultures our data agree with those of the RNAseq analysis, which also used ERM from monoxenic cultures. However, RiSMF2 expression was similar in IRM and ERM grown in the in vivo whole plant experimental system. These data have been discussed in the manuscript.
We agree that in Medicago (RNAseq data) RiSMF2 is the most highly expressed gene in the IRM, while in chicory roots (qRT-PCR data) RiSMF1 is the gene that shows the highest expression levels in the IRM. To us, this means that expression of these genes in the IRM depends on the host plant and/or experimental conditions, but not that our data are incorrect. We are very confident with these data since we have performed these gene expression analyses in three independent biological samples from three replicate experiments and we performed three technical replicates.
8. In the Discussion part (Figure 8), after the revision, the authors provided a schematic diagram of the working model of Mn and Fe transporters such as RiSMFs in different structures of R. irregularis, but it is not conclusive and not clear! First, these SMFs are not demonstrated to localize to PM. Second, RiSMF2 is mainly expressed in IRM, but not shown in this model.
We have simplified the diagram in order to make it clearer and have explained it in the discussion.
Finally, we would like to thank one more time this reviewer for all his comments, which indeed have improved the quality of our manuscript.